5

# Constraining on the Stationarity of Signal with Time-Frequency Surrogates to Enhance the Reliability of Singularity Spectrum Attributes of Random Seismic Noise Wavefield

Amir Ali Hamed<sup>1</sup>, Hiroe Miyake<sup>2,3</sup>, Zaher Hossein Shomali<sup>4,5</sup>, and Ali Moradi<sup>6</sup>

<sup>1</sup> Institute of Geophysics, The University of Tehran, 14155-6466, Tehran, Iran <u>amiralihamed@ut.ac.ir</u>.

<sup>2</sup> Earthquake Research Institute, The University of Tokyo, Tokyo, Japan. <u>hiroe@eri.u-tokyo.ac.jp</u>.

<sup>3</sup> Centre for Integrated Disaster Information Research, Interfaculty Initiative in Information Studies, The University of Tokyo, Tokyo, Japan.

<sup>4</sup> Institute of Geophysics, The University of Tehran, 14155-6466, Tehran, Iran <u>shomali@ut.ac.ir</u>.

<sup>5</sup> Department of Earth Sciences, The Uppsala University, 75236 Uppsala, Sweden; <u>Hossein.Shomali@geo.uu.se</u>.
 <sup>6</sup> Institute of Geophysics, The University of Tehran, 14155-6466, Tehran, Iran asmoradi@ut.ac.ir.

Correspondence to: Amir Ali Hamed (amiralihamed@ut.ac.ir)

Abstract. Existence of a self-affine long range persistence in the seismic noise time series evidences that the current state of
 system is not in the pure diffused regime and transition from coherent to incoherent motion is still on progress. Rate of this
 evolving transition can be indirectly linked to the degree of heterogeneity of medium, thus in this paper we tried to gain an

- insight into the heterogeneity of the medium by analyzing the width, extreme and asymmetrical trend of multifractal spectrum of seismic records. Nonetheless, toward high frequency ranges a seismic signal itself loses its stationarity short while after its recording onset time. Experimentally, the long-range correlation of a stationary time series (with 0 < h(2) <
- 1) can be discerned from a non-stationary process (with h(2) > 1) by examining the values of scaling exponent h(2), however, changing in the fractal properties in the crossover time scales in time series don't permit us to ascribe a single amount for h(2) and without executing additional analysis on the stationarity length of signals, direct calculation of such long range correlation and fractal dimensions might be biased. Hence, in this paper we examined the inherent stationarity of a signal relative to the different observation scales in the stochastic contexts before feeding the signal into the cycle of DFA.
- 25 This method is based on the comparison between global and local features of the original signal and its synthesized timefrequency surrogates; therefor it can effectively improve the accuracy of results. Our approach proves the existence of a high-velocity anomalous feature in the right flank of Sahand inactive volcano where it is surrounded by heterogeneous lowvelocity structures and extended to the shallower than ~3 km depth beneath this region at the northwestern of Iran.

Key words: Seismic Diffusivity, Multifractal Detrended Fluctuation Analysis, Seismic Stationarity, Statistical Noise Process

## 30 1 Introduction

Typically, each seismic wavefield is composed of two different coherent and incoherent parts, which may also be termed as the ballistic and the scattered waves, respectively. Ballistic waves carry mostly information about the signature of its

generating source and an ample image of the subsurface, nonetheless, trapped incoherent waves, arisen from randomly distributed scatterers or from a finite medium with an irregular bounding surface, due to the repeatedly sampling of medium, can be used in both imaging and monitoring purposes (*Snieder*, 2006). For propagation distances typically less than the mean free path, defined as the distance between two successive scattering anomalies, the coherent part is dominance over the

- incoherent part, while for times more than the mean free time, waves bounce on several heterogeneities and enter into the multiple scattering regime (*Hillers et al.*, 2012, 2013; *Galluzzo et al.*, 2015). Thereafter scattering sequences gradually distort the phase and amplitude of the incident plane wave front and also gain stability by temporal stabilization of the S-to-P deformation energy ratio, known as equipartitioning (*Hillers et al.*, 2012). For a weaker heterogeneous medium, the propagating wavefield will have the longer scattering mean free path or equally longer mean-free time, so a subtle difference
- in the mean-free time of wavefields indirectly provides the detailed knowledge about the heterogeneity level of the medium that they passed through. Since the emergence of seismic interferometry, about two decades ago, reconstructing Green's response of earth by the aids of equipartitioned source of energy has increasingly received much attentions with the seismological communities (*Campillo and Paul*, 2003; *Shapiro and Campillo* 2004). To come up with another fresh solution for learning the geological features of subsurface of earth, *Caserta et al.*, (2007) looked at this issue with different prospects.
- They utilized the concept of fractality to monitor the evolutionary motion of soil's particles against incoherent scattered excitations and compared it with a random walk Brownian motion. Leverage on such process they found that the average squared displacement of soil motion has an anomalous scaling-invariance feature i.e. a long range persistence fractional Brownian motion (*Sornette*, 2006). *Padhy* (2016) demonstrated that the coherent phase of seismic signals possesses the multifractal nature and the degree of multiractality toward higher frequencies and scattered phases decreases. This
- multiractality also evidences by *Matcharashvili et al.*, (2012) who explored the behavior of ambient noise in a calm day and the day perturbed by occurrence of earthquakes. They concluded that processes related to the preparation of local earthquakes generates coherent foreshocks therefore, they have a great influence on the multiractality of noise wavefield. *Telesca et al.*, (2015) studied the multifractal spectra of the seismic ambient noise during eruption of volcano and they found out this multifractality almost identical and much wider than that of the signal frame measured before the onset of the
- eruption. The capability of the Fractal Fluctuation Analysis (FFA) for discerning the incoherent wavefield from coherent part opened up a new road for *Pilz and Parolai* (2014) to investigate the fractality essence of particle motion in the near surface soil materials against random stationary excitation and they analyzed the spatial variation of mean free time of medium by determining the crossover between anomalous ballistic motion and normal diffusion.
- In all of these studies, the background seismic noise itself is generally assumed to be a locally reproducible quasi-periodic
  - and temporarily-stationary process with certain second-order statistical properties e.g., mean value, variance and autocorrelation functions, at all moments and over different realizations (*Bormann and Wielandt*, 2002) and other source of non-stationarity are mostly viewed as superimposed transient effects e.g. earthquakes. Multifractal Detrended Fluctuation Analysis (MF-DFA) is recently suggested to alleviate the effects of these non-stationarities, however, this method can just get a better handle on the detrimental effects of simple types of intermittent non-stationarities, associating with exterior long

period trends, e.g. polynomial, sinusoidal, and power-law functions (*Kantelhardt et al.*, 2002; *Kantelhardt*, 2012). Nonetheless, recent findings indicate that sometimes seismic wavefield itself loses its stationarity within short time intervals. This interval, in the long period bandwidth (0.1-1 Hz) and high frequency range (1-4 Hz), might be in the order of 1–1.5 h and 3–24 s long, respectively (*Gorbatikov and Stepanova*, 2008; *Wang et al.*, 2014; *Zhong et al.*, 2015a, b). This characteristic is substantially important. When the time series is stationary its behavior can be considered as a fractal

- Gaussian noise (fGn), while for a no-stationary signal the concept of fractal Brownian motion (fBn) should be used (*Qian*, 2003; *Ge and Leung*, 2013). So for the reliable detection of correlations, existing non-stationary of seismic signals can cause a misleading interpretation about its fractality, therefore, the effect of non-stationarity and fractality of time series should be carefully analyzed (*Chen et al.*, 2002; *Eke, et al.*, 2002; *Movahed et al.*, 2006). In this paper, we have geared toward to
- quantify statistical significance of stationarity by comparing its local and global features, e.g. time-varying spectrum (*Borgnant et al.*, 2010). By means of this approach, we can quantitatively decide about the stationary interval of signals before feeding them into the cycle of MF-DFA process.

The structure of this paper is organized as follows. At first we will describe the underlying rationale behind the measuring the degree of mutifractality of a time series by MF-DFA. Then, we will delineate how the stationarity of signal can be

assessed with time-frequency surrogates (*Xiao et al.,* 2007; *Borgnat et al.,* 2010). Finally, we will show how these observations provide compelling evidence for a direct causal link between diffusivity of wavefield and the degree of heterogeneity of medium. Thereby, the obtained results allow developing a deeper understanding of the mechanisms of noise wavefield in our studying region.

## 2 Datasets and Region of Study

- The study area, shown in Fig. (1), is located on the Turkish–Iranian plateau where 15-20 mm/year ongoing NNW oblique motion of the Arabian plate relative to Eurasian plate is accommodated through slip partitioning on the WNW-ESE striking right lateral strike slip faults (e.g., the North Tabriz Fault, NTF) and also by the thrust faults (*Jackson*, 1992; *McClusky et al.*, 2000; 2003; *McQuarrie et al.*, 2003; *Vernant* 2004; *Copley and Jackson*, 2006; *Reilinger et al.*, 2006; *Djamour et al.*, 2011). Fig. (1) shows the simplified geological map of northwestern of Iran, modified from *Solaymani et al.*, (2011). Existence of
- Marine limestones and marls of the Lower Miocene Tethyan Qom Formation in the northwestern and east of Tabriz city and on the islands of Lake Urmia indicates that the present day Tabriz Basin was part of the Tethys up to the Early Miocene (*Rieben*, 1935; *Stöcklin*, 1977; *Davoudzadeh et al.*, 1997), however, subsequent regression of the Qom Sea was followed by a period of continental sedimentation in northwestern and central Iran and Azerbaijan happened, by then the North Tabriz Fault (NTF) led to the formation of the Tabriz Basin at the Late Miocene (*Kelts and Shahrabi*, 1986). *Azad et al.*, (2015)
- determined the amount of maximum Pliocene-Quaternary cumulative dextral offset range from ~300 m and ~800 m for the east and west central sectors of Tabriz fault, respectively. They concluded that the horizontal component of displacements along the NW and SE segments of the NTF are absorbed within the Mishu and Bozghush ranges as two terminations of fault (Fig. (1)). We tend to understand the difference of shallow structure in the vicinity of NTF by comparing multifractal

5

characteristics of recorded seismic data. Noise records have been recorded by 1 broad-band (TABZ) and 8 short period stations of Iranian Seismic Telemetry Network (ISTN) data, operated by the Iranian Seismological Center (IRSC, *irsc.ut.ac.ir*). We removed the mean and the trend of all different-length fragments of seismic time-series, obtained from raw Nanometrics Y-File and GCF format datasets. Thereafter, they were corrected to the ground displacement by removing instrumental response. At last, these different-length fragments are merged for all three-component. All data-logger-induced spikes and glitches zeroed out by the Seismic Analysis Code. To handle the effects of gaps, stemmed from the zeroed out spikes and overlaps, we tried to fill them with linearly interpolated values with the help of free software package ObsPy (*Krischer, et al.*, 2015).

### **3** Methodology

#### 10 **3.1 Learning the fractality of signal by MF-DFA**

Many complex systems are composed of constituents that mutually interact in a complex fashion and the memory of this evolution is registered in the recorded time series. This evolving dynamic is assigned to a persistence evolution if large succeeding values of time series tend to follow the large preceding ones and the small values tend to follow small ones. This contrasts with anti-persistence time series, where large values of a time series tend to follow small ones. Our objective from

- fractality analysis of a complex process is to identify the presence of scale-invariance of fluctuations between nearly neighboring and distantly values. If the existence of a long-term persistence fractality, F, in a feature (e.g. a fluctuation or spectral power) is verified, the behavior of that feature can be described in accordance with a power law equation as:  $F(n) \sim n^h$ , in which F means the assigned feature, and n and h are the scale parameter and the scaling exponent, respectively. In order to detect the fractality of time series, several tools have been developed, among which MF-DFA can effectively
- differentiate the true fractal dynamics from fake scaling behavior by removing polynomial trends in the data. The main structure of this method is as follows (*Kantelhardt et al.*, 2002; *Kantelhardt*, 2012): *Step1*. We defined the common length for three components  $\xi = \mathbf{x}_1, \mathbf{x}_2, \mathbf{x}_3$ . Suppose  $u_i^{\xi}, i = 1, ..., N$  and  $Y^{\xi}(j)$  show respectively these time series and integration of them as

$$Y^{\xi}(j) = \sum_{k=1}^{J} u_k^{\xi}, \quad j = 1, 2, \dots, N , \xi = \mathbf{x}_1, \mathbf{x}_2, \mathbf{x}_3$$
(1)

This integration is also called the profile. Then, this profile is divided into the  $N_n = int(N/n)$  non-overlapping segments of length *n*. Since the record length will not always be an integer multiple of *n*, a short part at the end of the time-series remains in the most cases, which we have ignored them from the cycle of MF-DFA. *Step 2*. Detrending step is done through estimating a polynomial of order  $\ell$  for each segment by least square fitting. *Step 3*. This trend is subtracted from the original profile, so that the residual sequence for each segment  $k, k = 1, ..., N_s$ , can be obtained as

$$F_{\xi}^{2}(n,k) = \frac{1}{n} \sum_{i=1}^{n} \{Y^{\xi}[(k-1)n+i] - y_{k}(i)\}^{2}$$
 (2)

where  $y_k(i)$  indicates the fitting polynomial in segment k and the qth order overall detrended fluctuation is calculated by

averaging over all segments and also the results for three components,

$$F_q(n) = \left(\frac{1}{N_n} \sum_{\xi = \mathbf{x}_1, \mathbf{x}_2, \mathbf{x}_3} \sum_{k=1}^{N_n} \left(F_{\xi}^2(n, k)\right)^{q/2}\right)^{1/q}; \quad F_q(n) \sim n^{h(q)}$$
(3)

where h(q) is the generalized Hurst index, but for q = 0 this equation is revised as

$$F_0(n) = \exp\left\{ \left( \frac{1}{N_n} \sum_{\xi = \mathbf{x}_1, \mathbf{x}_2, \mathbf{x}_3} \sum_{k=1}^{N_n} \ln[F^2(n, k)] \right) \right\}$$
(4)

- If we plot  $\log_2(F_{10}(h))$  versus  $\log_{10}(h)$ , a straight line with the slope h(q) for a given value of q is obtained. The slope h(q), for positive values of q, represents the scaling behavior of segments with large fluctuations, while for negative values of q, it describes the scaling behavior with small fluctuations. Therefore, for a mono-fractal time series, the achieved value for h(q) will be a unique amount, independent to the values of q. Using the Legendre transformation we can define the multi-fractal spectrum equation  $D_q(h) = qh(q) t(q)$  in which t(q) = qh(q) 1 and h(q) = dt(q)/dq are respectively
- the q-order mass exponents and the q-order generalized Hurst index. While, for a mono-fractal series, h(q) is independent of q, D(q) depends still on the q. In that case, t(q) will have a linear dependence to q and the width of its multi-fractal spectrum will zero length, otherwise the t(q) will depend on the q through a non-linear relationship and multi-fractal spectrum will be broader in the shape. By fitting a quadratic function

$$D_q(h) = A[h(q) - h_0(q)]^2 + B[h(q) - h_0(q)] + C$$
(5)

around the maxima of spectrum,  $h_0(q)$ , *Shimizu et al.*, (2002) proposed an excellent tool for experimentally visualizing the shape of multi-fractal spectrum. By extrapolating curves to zero, the width of the spectrum can be defined as

$$W = h_{max} - h_{min}$$
; with  $D_q(h_{max}) = D_q(h_{min}) = 0$  (6)

where the width of the spectrum is zero for a mono-fractal series (*Shimizu et al.*, 2002; *Telesca et al.*, 2004; *Telesca et al.*, 2015). For q = 1,  $D_q(h)$  can be interpreted as a fractal dimension, while q = 2 and q = 3 represents the information

- dimension and the correlation dimension of a time series, respectively. The parameter B returns the asymmetrical trend of spectrum. In particular, a left tail spectrum represents that the dynamics of time series is controlled by the large transient fluctuations, with positive values of q, while a right tail spectrum indicates the relative dominance of small smoothly varying fluctuations, with negative values of q (*Telesca et al.*, 2004). Hence, when spectrum has a long left tail shape it can be inferred that scattering is pronounced in the region, as a result, the complexity of a multifractal spectrum may provide a
- crude insight into the heterogeneity of medium. This complexity can be characterized in terms of width and shape of multifractal spectrum by three parameters  $(W, h_0, B)$  (*Padhy*, 2016).

### **3.2 Testing Stationary of Signal**

As highlighted above, seismic noises can be described as superposition of signals emitted by numerous independent, timevarying and spatially distributed sources. This conception indicates that the stationarity of seismic noises might highly be

15

questionable and it should be surveyed more precisely. When it comes to dealing with the actual implantation, for a stationary time series, the value of scaling exponent h(q) at q = 2 is equal to that of the Hurst exponent H and its value will be at the range of 0 

5

25

created directly from original data. The latest one has a comparative advantage because they do not depend on a particular model, nor on any parameters, thus they are non-parametric in nature (*Theiler et al.*, 1992). In this approach surrogate data are created by means of Fourier Transform through which the magnitude of the spectrum of original signal is preserved and then the organized phase structure, which controls the non-stationarity of signal, is destroyed with new (uniformly random) phases  $\phi(f)$  over  $[-\pi,\pi]$ . To construct the surrogates we have to use the inverse Fourier transform with the symmetrized phases,  $\phi(f) = -\phi(-f)$ . It makes signal a real quantity with no imaginary components. Once a collection of *J* synthesized

surrogates,  $\{s_j(t), j = 1, ..., J\}$ , are generated, dissimilarity between local  $w_x^S(t, f)$  and global spectra,  $w_x^{avS}(t, f)$ , for both original signal and its surrogates can be evaluated (*Borgnant et al.*, 2010):

$$\left\{c_n^x = D\left(w_x^S(t, f), w_x^{avS}(t, f)\right), t = 0, \dots, T\right\}$$

and
$$\left\{c_n^{s_j} = D\left(w_{s_j}^S(t, f), w_{s_j}^{avs}(t, f)\right), t = 0, ..., T, j = 1, ..., J\right\}$$
 (11)

where  $c_n^x$  and  $c_n^{s_j}$  are the dissimilarities of original and surrogate signals, respectively. Statistical variance  $\Theta_1 = var(c_n^x)_{n=1,\dots,N}$  gives a measure of how the spectra for original data distributes itself about the expected value. Similarly, for each one of J synthesized surrogates it can be measured variances as

$$\left\{\Theta_0(j) = \operatorname{var}\left(c_n^{s_j}\right)_{n=1,\dots,N}, j=1,\dots,J\right\}$$
(12)

As a null hypothesis original signals is supposed to be stationary but if it violates the predefined threshold  $\gamma$ , null hypothesis is rejected and non-stationarity is assumed

$$d(x) = \begin{cases} 1 & if \quad \Theta_1 > \gamma: non - stationarity \\ 0 & if \quad \Theta_1 

By inserting Eq. (15) into the Eq. (11), we can define  $\theta_1$  and also a set of outputs for  $\theta_0$ . To obtain a threshold for  $\gamma$ , these  $\theta_0$ s can assume as a set of realizations for a given probability distribution e.g. Gamma distribution

$$P(x; a, b) = \frac{1}{b^a \psi(a)} x^{a-1} \exp\left(-\frac{x}{b}\right)$$
(18)

This probability distribution enables us to define the mean of realization, that is  $\mu$ . Moreover, the threshold value for  $\gamma$  is also 5 considered as a confidence level of 95% for probability distribution under the maximum likelihood sense. By comparing  $\theta_1$ and the estimates of  $\theta_0$ , one can define the degree of stationarity. Quantitatively, these difference can be evaluated by index of non-stationarity (INS) (*Xiao et al.*, 2007):

$$INS = \sqrt{\frac{\theta_1}{\frac{1}{J} \sum_{n=1}^{J} \theta_0(j)}}$$
(19)

Further, note the result of stationarity test depends on the window length of spectrogram,  $T_n$ . This dependence can be analyzed by the scale of non-stationarity (SNS). It informs us that in the which one/ones of considered values for  $T_n$  the given threshold in Eq. (13) has been exceeded (*Xiao et al., 2007*):

$$SNS = \frac{1}{\tau} \arg \max_{T_n} \{INS(T_n)\}$$
(20)

### 4 Results

We retrieved the average of three components for one-hour-length displacement signals at all stations. Signals have been recorded at 50 sample per second so each one of these signals has  $N = 3600 \times 50 = 180000$  samples. Fig. (2a) shows a typical example of this average and its spectrum recorded by TBZ station. The magnified part of this spectrogram at the 100-200 s interval of this time series (Fig. (2b)), clearly verifies the existence of non-stationary signals at the 1-4 Hz, but without using Time-Frequency analysis we cannot confidently decide about the length of stationarity in this interval (*Wang*, 2014; *Zhong*, 2015). Therefore, before going further, the degree of stationarity of signal should be searched in advance. In the

- range of high-frequency 1-4 Hz transient signals are dominant, while the story is completely different at 0.3-1 Hz bandwidth and the presence of higher mode surface waves might increase the probability of seeing a stationary source of energy. Therefore, we should analyse the stationarity of signal within the separate 1-4 Hz and 0.3-1 Hz frequency intervals. To make the process of fractality more trustworthy, the length of signal should be chosen in compatible with the length of stationarity of signal at the 1-4 Hz, before feeding it into the MF-DFA process. At the same time, the length of this curtailed window
- have to be long enough that the longer period of signal remains in its macroscale stationarity interval. To calculate the local spectrogram in the Eq.(7) we used just the first six Hermite functions and also the local window length of  $T_n = 0.1 s$ . We used the Eq.(12) for calculating the empirical distribution for  $\theta_0$  and its corresponding fitted thick blue curve is shown in the Fig. (2c). Empirical distribution is achieved based on J = 500 surrogates. While the red line denotes the statistic  $\theta_1$  of the time series, the thin vertical dashed black line shows the upper 5% quantile of this distribution and it is used as the threshold
- for stationarity limit of signal. Within this window length, the signal evidently has lost its stationarity in the range of 1-4 Hz, while it confidently passed the test in the range of 0.3-1 Hz (Fig. (2c)). We changed the range of local spectrogram window

25

length  $T_n$  from 0.01 to the 0.45 of the length of the time series. As aforementioned, a noise time series will be stationary if it can pass the stationarity test under all of these  $T_n$  values. Comparing the INS between these results provides a physical interpretation in this regard and helps us in choosing the reliable interval of stationarity. Clearly, INS values, denoted by the dashed star red line, is more or less locates in the higher than the thresholds, shown by the dashed star blue line, at the 1-4 Hz

- 5 and this evidences that the degree of stationarity of signal in this range of frequency have to be explored in the less than 100 s time window. Our subtle searches show that in this window, the length of stationarity of signals at the 1-4 Hz bandwidth varies between 1-36 s and this interval is comparable with the results of *Gorbatikov and Stepanova* (2008). The 36 second length of signal, shown by two-sided white arrow in the Fig. (2b), passed the stationarity test and coincidently satisfies the macroscale stationarity interval for the whole range of 0.3-4 Hz frequency bandwidth (Fig. (2d)), so we have separated this
- stationary part of signal to the MF-DFA. It should be noticed that a linear filter has no significant effect on the scaling properties of signals (*Chen* 2005, *Pilz and Parolai*, 2014). Similar approach has been employed for choosing the suitable length of stationarity for other stations. As the next step, we have to decide about the suitable segments h for executing the MF-DFA. To increase the reliability of this process, we adapted the same strategy as *Pilz and Parolai*, (2014), by limiting the size of segments into the  $h \le N/6$  samples, that is,  $h \le 6 s$ . This constrain ensures us that the number of segments for
- averaging procedure is always sufficient since from statistical point of view violating the upper N/6 threshold may cause the unreliable results (*Pilz and Parolai*, 2014). We have also removed a second order polynomial in the detrending step. Generally,  $\log_{10}(F_2(h))$ - $\log_{10}(h)$  grows as a power-law relationship respect to the *n* and we expect this increasing gives a feedback about the presence of scaling behavior in the signal, however, toward the samples smaller than (10/sampling rate), the deviation from a straight line in the log-log plot of power law equation might give rise to a
- 20 misleading interpretation so to compensate this effect, we have to modify the  $F_q(h)$  by the help of the shuffled fluctuation  $F_q^{shuf}(h)$  (Kantelhardt et al. 2002, Movahed et al., 2006)

$$F_q^{mod}(h) = F_q(h) \times \left( \langle \left[ F_q^{shuf}(\hat{h}) \right]^2 \rangle^{1/2} h^{1/2} / \langle \left[ F_q^{shuf}(h) \right]^2 \rangle^{1/2} \hat{h}^{1/2} \right) \qquad \left( for \, \hat{h} \gg 1 \right)$$
(21)

where  $\langle [F_q^{shuf}(\hat{h})]^2 \rangle^{1/2}$  denotes the usual MF-DFA fluctuation function, averaged over several configurations of shuffled data and  $\hat{h} = N/40$ . These shuffled data are taken from the original time series, in which the data are put into a random order (*Kantelhardt et al.* 2002).

Fig. (3a) shows the plot of fluctuation functions,  $\log_{10}(F_q(h))$ , versus  $\log_{10}(h)$  (in second) for the 36 s data at the TBZ station with 1-step incremented q values, ranging from -8 to 8. By taking a closer look to the slope of curves we can see that they are changed along the different intervals, but this changes are not identical for all values of q, so this evidences the multifractality of signal. We expect that the existence of different long-term correlations for small and large fluctuations in

30 the noise record are most likely the reason of the observed multifractality, nonetheless it should be noted that a broad probability distribution can also produce the multifractality, so we should trace the origin of this multifractality and distinguish the type of multifractality in the time series by analyzing the corresponding randomly shuffled series. By

5

randomly shuffling strategy, any correlations due to the order of the successive samples of a time series will be destroyed, otherwise it can be deduced that multifractality stems from a broad probability distribution. Fig. (3b) gives a clear comparison between the plot of h(q) versus q for TBZ station and its corresponding randomly shuffled time series. The shuffled series exhibits a weaker scaling behavior, so the multifractality due to correlation is dominant, although the multifractality due to a broad probability distribution cannot completely be ignored. The same pattern has been observed for

other stations that are not shown here.

More likely the broadness of multi-fractal spectrum and a long right-skewed trends of these spectrums, given in Table 1, indicates that dynamic of signal is mostly controlled by the existence of small converted and scattered fluctuations, arisen from the heterogeneous structure beneath of those stations, so comparing the multifractal spectrum of all stations can provide

- 10 an ample view of heterogeneity at the studying region (*Padhy*, 2016). Recent studies e.g. *Chiu et al.* (2013) declares that the origin of obtained samples at the east part of Lake Urumieh dates back to the Late Miocene (11 Ma), but toward Sahand and Sabalan post-collisional volcanoes there are evidences of the younger eruptions, corresponding to the Late Miocene to Pliocene (6.5-4.2 Ma) and Quaternary (