# Peer review of "Constraining on the Stationarity of Signal with Time-Frequency Surrogates to Enhance the Reliability of Singularity Spectrum Attributes of Random Seismic Noise Wavefield"

_Nonlinear Processes in Geophysics, 2017_

## Referee Comment (RC1) · Anonymous Referee #1 · 15 Jul 2017

Abstract: 1) the authors state "Existence of a self-affine long range persistence in the seismic noise time series evidences that the current state of system is not in the pure diffused regime and transition from coherent to incoherent motion is still on progress" however not in the Results section nor in the Disussion section it was ever strengthened such statement on the base of the obtained results, leaving it sospended and without a clear connection with all the performed analysis. 2) It is not clear (in the worst case, not correct) that the long-range correlation of a stationary time series (with $0 < h(2) < 1$) can be discerned from a non-stationary process. Stationarity and non-stationarity are

different characteristics and such relationship that makes one to be discerned from the other is obscure. 3) The authors say "changing in the fractal properties in the crossover time scales in time series don't permit us to ascribe a single amount for h(2)", probably the authors link the multifractality to the crossover time scale in the fluctuation functions. This is not what multifractality means. The crossover in a fluctuation function indicates just the co-existence of two different dynamics at different time scale ranges. However, in the paper no crossover timesclaes have been mentioned; so such sentence does not reflect any aspect of the topic approached in the paper. 4) Most important: the authors claim that their pre-processing (related with the identification of stationary segments) improve the accuracy of the results. But, no comparison with other methods has been provided, to assess that their pre-processing not only is original but also effectively improves the accurary of the results.

Introduction: 1) It is not clear the concern of the authors in selecting the stationary intervals of signals before using the MFDFA, if the MFDFA is already capable to deal with nonstationarities. Moreover, the authors have not clarified, or not mentioned at all, what type of nonstationarities would affect their data, so that the application of the MFDFA directly would produce misleading results. If the nonstationarities of their data are among those types that MFDFA would be able to deal with, why the pre-processing is proposed in the next sections?

Methodology 1) The title of subsection 3.1 seems not appropriate, since the fractality of a signal can be detected or identified and not learned 2) The authors say that "a short part at the end of the time-series remains in the most cases, which we have ignored them from the cycle of MF-DFA", practically they do not perform the MFDFA in the reverse side of the time series in order to not disregard the last part of the time series that is necessarily left out. This is not well-done procedure, because even the last part of the signal could contain information, in principle, that could be useful for the overall calculation of the fluctuation function. Actually, you don't know if including such small part that is left at the end of time series would change significantly your results or not.

So, it would be much better to apply the MFDFA a sit was proposed by several authors, so, forward and backward. 3) It is obscure at all, the summation the authors did in eq. 3; practically they sum over the three different time series, so instead to consider one set of fluctuation functions (depending on q) for each time series, they summed for each q the three fluctuation functions obtained for each time series. I suppose that each time series refers to one direction of the sensor by which the seismic noise was measured, one vertical and two horizontal. Actually, it would has been much more useful and informative of the underlying geophsyical process to analyse each direction separately. Probably it would has been better to first calculate the total displacement combining the three time series and then apply the MFDFA on such total displacement. 4) Eq (6) is not correct, because the zero-values are of the parabolic fitting function and not of Dq, calculated from the data. 5) At page 7, the authors say that the "phase structure, which controls the non-stationarity..." this is not correct, because the phase are responsible of the non-linearity of a time series. The stationarity/nonstationarity of a time series can be simply verified looking at the power spectrum and its power-law shape, which depends solely on the amplitude of the Fourier transform and not on the phase. 6) The Eq. 10 seems misleading if compared with the analogous Eq. 3 of Borgnat et al. (2010), because the multitaper spectrogram that the authors use is evaluated only at time positions tn, with n=1,..N, with a spacing which is an adjustable fraction—typically, one half—of the temporal width Th of the K windows Hk(t), which are the Hermite functions. In Eq. 10, the authors use to vary a continuous parameter t from 0 to T. So, in the eq. 3 of Borgnat et al. (2010) the averaged spectrum is done on N local spectra. It is not clear on how many local spectra the average is done in Eq. 10 of the present paper. 7) Eq. 15 seems not correct (compared with eq. 16 of Borgnat et al. (2010)) 8) At page 8, the authors mention the parameter Tn as the window length of spectrogram, but they never used or defined it previously. 9) The equation (20) is introduced but never used.

Results 1) At page 9, N/6 samples correspond to 600 seconds and not 6 seconds. 2) The authors mention that they calculated the modified Fq to compensate the effect at small scales, but at the end they showed in Fig. 3a the un-modified Fq. 3) Fig. 3a shows that the fluctuation functions by approximately two differente regimes, at small scales until approximately and at long scales from 1 until the end of the investigated timescale range. The authors do not do any comment on such apparent double regimes and seems that they have calculated the scope of the lines fittings each fluctuation function in the entire range of scales (actually, the authors have not clarified/specified if they used the entire scale range, or part of it). If they used the entire range, the results obtained about the multufractality are absolutely wrong. 4) It is lacking a convincing geophysical explanation of the link between the tectonics of the area and the found multifractal paramters.

---

## Editor Comment (EC1) · L. Telesca (Editor) · 6 Oct 2017

The paper presents several flaws in the organization, in the methodological approach and in the presentation of the results. Unfortunately the English is not fluent and the clarity of the sentences and concepts is not always achieved. Even the abstract, which in principle has to outline clearly and synthetically the main findings of the study, seems quite obscure and does not convey clearly the information on what is the outcome of the presented research. For instance the starting paragraph of the Abstract "Existence of a self-affine long range persistence in the seismic noise time series evidences that the

current state of system is not in the pure diffused regime and transition from coherent to incoherent motion is still on progress. Rate of this evolving transition can be indirectly linked to the degree of heterogeneity of medium. . ..." seems not well explained and it would be difficult for a reader to understand what is exactly its meaning. Please, be aware that at least the abstract should be developed in a manner that even a reader not strictly familiar with the topic of the paper can capture the general information. Unfortunately, the whole abstract fails in the characteristics of clarity, synthesis, clear explanation of the obtained results.

Such lack of clarity is also evidenced in the description of the dataset. It is not mentioned how many stations have been analysed, although one can guess them from Fig. 1; but probably an explanatory table indicating name, geographic coordinates, elevation, and maybe some simple statistical characteristics, would have been useful to add to make the text clearer. However, the authors say that after removing mean and trend (which trend? linear trend? a figure with the raw data would have been useful), they merged all the different length segments; but how such merging was performed? Then since the data present gaps "stemmed from the zeroed out spikes and overlaps" (what overlaps?), this gaps were filled with linear interpolation; but this interpolation is not clearly explained, and the number and the length of gaps is not specified: these details would be important to mention especially in a journal like NPG, where a relevant focus is given on the methodological aspect of presented study.

Some flaws also exist in the methodology. For instance it would have been more correct to link the persistence/antipersistence of a signal to the succession of the increments rather than of the signal values. It is correct the observation of the referee about what the authors did, ignoring the small part of the signal at its end that remains out during the calculation of the fluctuation function, since in most of the studies such small part at the end of the signal has not ignored but included recalculating the fluctuation function starting from the end of the signal. Also the use of the multitaper spectrogram (Borgnat et al., 2010) seems not correctly performed or at least not clearly carried out, raising

issues on the correctness of the obtained results.

The authors apply a complex signal pre-processing for searching the stationary windows to apply MFDFA. Besides the logical observation of the referee that the algorithm of the MFDFA is already developed in a way to remove the non-stationarities (thus making probably quite unuseful or unnecessary that pre-processing), it would have been, instead, much more useful, to apply the MFDFA directly to the signals (as obtained after the procedure described in section 2) and then to such stationary segments (and thus, after the pre-processing) to check if any difference would have been existed and to see if an improvement would have been obtained in the results, especially in relationship with the geophysical implications. I am also skeptical about the obtained results, because it seems that the calculation of the slopes of the fluctuation functions in Fig. 3 was performed considering all the available shown scales; if so, this is clearly wrong, because the fluctuation functions for any q are not linear in log-log scales. So, if the geophysical interpretation of the results are based on such wrong calculations of the slopes of the fluctuation functions, also all the geophysical implications, rather poorly described by the way, would be not convincing.
* * *

---

## Author Comment (AC1) · 8 Jan 2018

EC: Unfortunately the English is not fluent and the clarity of the sentences and concepts is not always achieved. Even the abstract, which in principle has to outline clearly and synthetically the main findings of the study, seems quite obscure and does not convey clearly the information on what is the outcome of the presented research.

AC: Many thanks for the comment. The abstract contents were thoroughly revised as follows:

[Figure]

Extended Abstract. The diffusivity of incoming seismic noise is certainly a critical pre-condition for executing seismic interferometry. But higher than the narrow $\sim$ (0.05 -0.3) Hz microseismic bandwidth, this diffusivity stems mostly from the heterogeneity of local site characteristics, therefore the heterogeneity level of sites should be assessed beforehand in order to make an accurate assessment of a Green's response. As evidenced by recent studies (e.g. Padhy 2016), it has become evident that seismic signals show a self-affine long-range persistence in their coherent parts (e.g. P or S body waves) which is slowly disappeared with the emergence of the incoherent diffused incoming wavefield (i.e. Coda waves). Pilz and Parolai (2014) showed that the rate of this evolving transition is closely linked to the heterogeneity level of a medium in such a way that for a strong heterogonous medium less time will be needed for falling signal into the diffuse state. Therefore learning the fractality of a seismic noise will indirectly provide the basis for a decision on the potential place for executing this method. But this conclusion rests on this pillar that input incoming noise wavefield is always stationary, but there is obviously a degree of ambiguity surrounding such assumption. There may be circumstances under which signals include: a) Intrinsic Non-Stationary Direct Waves (as indicated by Hillers and Ben-Zion, 2011, the interaction of wind and topography, by near-surface microcracks or micro earthquakes may repetitively generate a reproducible source of energy. Because of this repetitive reproducibility, they do not necessarily equivalent to temporary superposed perturbations. So depending on the generating source type and distance from the receiver, they might cause large variability in the characteristics of seismic noise signals.). b) Intrinsic Non-Stationary Scattered Waves (see, for instance Meng, et al., 2015 who showed that the incoherent coda waves induced by multiple scattering is a non-stationary signal, or Margerin et al., 2016 where they showed that stronger emphases need to be placed on the active scattering in an unstable perturbed medium), Intrinsic Stationary/ Non-Stationary Direct Waves). c) External non-Stationary signals (e.g. segments with different properties, random outliers or spikes with different amplitudes, etc.). In executing the fractal analysis, it is essential that the method chosen be consistently reliable to ensure that

the correct Hurst coefficient is being used for the interpretation. There is broad agreement on the appropriateness of MFDFA in studying multifractal scaling behavior of non-stationary time series, but it fails to comply with the intrinsic non-stationarity of signals (please take a look at the responses given to the RC1 comments, to find complete explanations in this regard.). In this paper, we used the method introduced by Borgnat et al., (2010) to recognize the inherent characteristics of signal in the pre-processing step, before the feeding data into the cycle of Fractal analysis. Based on this revised method we try to define the degree of heterogeneity of different sites, located at the North-Western of Iran.

References:

Borgnat, P., Flandrin, P., Honeine, P., Richard, C., and Xiao, J.: Testing stationarity with surrogates: A time-frequency approach, IEEE Transactions on Signal Processing, 58, 3459-3470, 2010.

Hillers, G., and Ben-Zion, Y.: Seasonal variations of observed noise amplitudes at 2–18 Hz in southern California, Geophysical Journal International, 184, 860-868, 2011.

Margerin, L., Planès, T., Mayor, J., and Calvet, M.: Sensitivity kernels for coda-wave interferometry and scattering tomography: theory and numerical evaluation in two-dimensional anisotropically scattering media, Geophysical Journal International, 204, 650-666, 2015.

Padhy, S.: The Multi-fractal Scaling Behavior of Seismograms Based on the Detrended Fluctuation Analysis, in: Fractal Solutions for Understanding Complex Systems in Earth Sciences, Springer, 99-115, 2016.

Pilz, M., and Parolai, S.: Statistical properties of the seismic noise field: influence of soil heterogeneities, Geophysical Journal International, 199, 430-440, 2014.

---

## Author Comment (AC2) · 9 Jan 2018

The author would indeed prefer to answer these three questions together. Q (1) It is not clear (in the worst case, not correct) that the long-range correlation of a stationary time series (with $0 < h(2) < 1$) can be discerned from a non-stationary process. Stationarity and non-stationarity are different characteristics and such relationship that makes one be discerned from the other is obscure. Q (2) Moreover, the authors have not clarified, or not mentioned at all, what type of nonstationarities would affect their data, so that the application of the MFDFA directly would produce misleading results. If the nonstationarities of their data are among those types that MFDFA would be able to deal with, why the pre-processing is proposed in the next sections? Q (3) It is not clear the concern of the authors in selecting the stationary intervals of signals before using the MFDFA, if the MFDFA is already capable to deal with nonstationarities. Moreover, the authors have not clarified, or not mentioned at all, what type of nonstationarities would affect their data, so that the application of the MFDFA directly would produce misleading results. If the nonstationarities of their data are among those types that MFDFA would be able to deal with, why the pre-processing is proposed in the next sections?

Answer: In general, there is broad agreement on the appropriateness of MFDFA in studying multifractal scaling behavior of non-stationary time series, but we wish to draw your attention to the paragraph "III. ANALYSIS OF SUNSPOT TIME SERIES" of Movahed (et al., 2005) which explains the reasons why further attention must be taken in the analyzing the stationarity of signals in the pre-processing step, before feeding them into the cycle of Fractal analysis. In connection with this point, we MUST make it absolutely clear that inherent non-stationarity of signals should never be confused with the concept of non-stationarity made by external perturbations. Regularly, fractional Gaussian noises (fGn) are involved in the inherent stationarity process, in contrast, fractional Brownian motions (fBm) are linked to the inherently non-stationarity process. The stationarity of fGn signals can be characterized by two parameters, $\sigma^2$, the variance, and H, the Hurst coefficient, while a fBm process has a time-dependent variance. Hence, on the basis of the class to which signals belong, different techniques may be required for processing. Failure to match signal class with the appropriate method of fractal analysis results in serious error in the estimating H. For instance Dispersional analysis (Disp) is recommended to the analysing the fractionality of fGn signals, while bridge detrended scaled windowed variance analysis (bdSWV) is suitable for fBm signals (Eke et al., 2000). These classes might not be a-priori known, so a preliminary interpretation in this regard may be available by fitting a straight line of slope $-\beta$ on a log-log plot of the periodogram. Based on this method, signals can be categorized according to the value of $\beta$, but Eke et al., (2000) placed emphasis on this point that this method is only

applicable if $\beta$ falls into the category -1<$\beta$<0.38 (for an obvious stationary case) or if falls into the range of 1.4 <$\beta$<3 (for an obvious non-stationary case), but there is no certainty that this method fully comply with the complicated characteristics of signals in the range of 3.8<$\beta$<1.4 where stationary and non-stationary mixed into each other. Based on this assumption, it is essential to provide another reliable framework for a regular monitoring of stationarity of signals. Signal summation conversion method (SSC) is advised to use as a discriminating method (Eke et al., 2000). An alternative approach for distinguishing fGn signals from fBm signal was proposed by Movahed et al., (2005) who experimentally attempted to examine the feature of Sunspot Time Series by analyzing the behavior of standard deviation of its time series as a function of timescale. There is a growing consensus amongst researchers (Zhong, et al., 2015; Wang et al., 2014) that Time-Frequency Surrogate Analysis (TFSA) proposed by Borgnat et al. (2010) can provide a complementary view for testing stationarity of seismic signals in an operational sense. This method is statistically characterized on the basis of a set of surrogates which all share the same average spectrum as the analyzed signal while being stationarized. With this short introduction, the answers to the above-mentioned question can be summarized briefly: We, therefore, agree with this statement in your question: Stationarity and non-stationarity are different characteristics, and this is exactly what we need. In other words, this difference enables us to discern a stationary fGn from non-stationary fBm. We want to underline the point that nonstationarities from our point of view are those corresponding to inherent non-stationarity which should be known at the pre-processing step before making our choice between fGn and fBm process. TFSA is a rigorous and sufficiently flexible method which not only provides an opportunity for assessing the inherent stationarity/non-stationarity of a signal but also further it may bring information from their state of stationarity. Therefore, another strength of TFSA lies in its capability to constrain the length of stationary of the signal by quickly and reliably validating tasks. This would be of utmost importance if our data encompass the microseismic range of frequency (0.1-0.3 Hz) or if data were acquired from the surveys at the vicinity of dome-looking topography features seismic signals

appear mostly in the quasi-stationary state. In those cases, reproducible seismic signals could fall into one of the following states: macroscale, mesoscale or microscale state (Fig 1), therefore, more accurate method is needed in order to properly assess the state of this quasi-stationarity. In this paper, we do make a point that the length of the signal directly impacts on the reliability of Long-range autocorrelations assessment. The importance of this factor was previously the subject of other investigations such as Delignieres et al., (2006) and Warlop et al., (2017), but this point has been hidden from view in analyzing the fractality of seismic signals. The importance of this issue is strikingly apparent for seismic signals since the stationarity of seismic records at various frequency ranges or temporal length are different. The experimental estimates obtained by Gorbatikov & Stepanova, (2008) shows that, at the microseismic range of frequency, signals are mostly quasi-stationarity, but this stationarity may not be preserved for very long periods of time. For instance, this interval might be lengthened to the several days or be shortened to the 1–1.5 h, while Wang et al., (2014) showed that for frequencies above than 1 Hz signals, the stationarity range of the signal is just in the range of several seconds. This is where we think we have to focus much of our attention. Obviously, by choosing an extreme short window length users might thereby misunderstand the true origin of non-stationarity, in such a way that lead to results that are non-informative, and potentially misleading. Accordingly, we will be able to adaptively adjust the length of processing incompatible with the stationarity length of the signal, if we take full benefit of the advantages of TFSA.

References [1] Bashan, A., Bartsch, R., Kantelhardt, J. W., & Havlin, S. (2008). Comparison of detrending methods for fluctuation analysis. Physica A: Statistical Mechanics and its Applications, 387(21), 5080-5090. [2] Borgnat, P., Flandrin, P., Honeine, P., Richard, C., & Xiao, J. (2010). Testing stationarity with surrogates: A time-frequency approach. IEEE Transactions on Signal Processing, 58(7), 3459-3470. Eke, A., Herman, P., Bassingthwaighte, J., Raymond, G., Percival, D., Cannon, M., et al. (2000). Physiological time series: distinguishing fractal noises from motions. Pflügers Archiv, 439(4), 403-415. [3] Movahed, M. S., Jafari, G., Ghasemi, F., Rahvar, S., & Tabar,

M. R. R. (2006). Multifractal detrended fluctuation analysis of sunspot time series. Journal of Statistical Mechanics: Theory and Experiment, 2006(02), P02003. [4] Wang, D., Li, Y., & Nie, P. (2014). A study on the Gaussianity and stationarity of the random noise in the seismic exploration. Journal of applied Geophysics, 109, 210-217. [5] Warlop, T., Bollens, B., Detrembleur, C., Stoquart, G., Lejeune, T., & Crevecoeur, F. (2017). Impact of series length on statistical precision and sensitivity of autocorrelation assessment in human locomotion. Human Movement Science, 55, 31-42. [6] Zhong, T., Li, Y., Wu, N., Nie, P., & Yang, B. (2015a). Statistical analysis of background noise in seismic prospecting. Geophysical Prospecting, 63(5), 1161-1174. [7] Zhong, T., Li, Y., Wu, N., Nie, P., & Yang, B. (2015b). Statistical properties of the random noise in seismic data. Journal of applied Geophysics, 118, 84-91.

===========================================================

Q (4) The authors state "Existence of a self-affine long-range persistence in the seismic noise time series evidence that the current state of system is not in the pure diffused regime and transition from coherent to incoherent motion is still on progress" however not in the Results section nor in the Discussion section it was ever strengthened such statement on the base of the obtained results, leaving it suspended and without a clear connection with all the performed analysis.

Answer: Generally speaking, a local perturbation may continuously fluctuate system over time in a complex manner, such that consecutive cycles of a signal exhibits an interdependency spanning over long time intervals. Indeed, the existence of this type of long-range correlation (or self-affine long-range persistence) in the seismic noise wavefield is mostly associated with the existence of coherent signals (e.g. P, S, and Surface waves). Matcharashvili, et al. (2013) showed that the dynamical features of ambient noise undergo essential changes during preparation, and also after triggering the activity of strong local events. At distance far away from a seismic source, the generated waves bounce on several heterogeneities and gradually enter in the multiple scattering regimes (see Fig. (2)). Therefore, the diffused scattered coda waves overwhelm the

direct wave, gradually. In this case, the strength of persistence of signal diminishes through time, due to the attenuation of the coherent wavefront. Therefore, the footprint of a perturbation should be tracked in the ambient noises and the existence of a self-affine long-range persistence in the seismic noise time series evidences that this system is still in the process of transition, that is, transition from a complete coherent state to a pure diffusive state. In this point of view, the scattering mean free time, $\tau$, indicates the characteristic time after which such transition is happing. The scattering mean free time is the key feature of a medium since it allows knowing in advance the degree of heterogeneity of that medium. A medium with low levels of heterogeneity is no longer considered as a candidate site for executing the advanced process such as seismic interferometry (see, for instance, see Wapenaar, 2012 a,b), since the existence of a diffusive wavefield is a key precondition for this analysis (Pilz & Parolai 2014). With this short introduction, the answers to the above-mentioned question can be summarized briefly: In short, this sentence "Existence of a self-affine long-range persistence in the seismic noise time series evidence that the current state of system is not in the pure diffused regime and transition from coherent to incoherent motion is still on progress" is evident from recent studies not directly the conclusion of our research but we MUST give the specific reference for this argue. Our paper is actually the continuation of the effort made by Pilz & Parolai (2014), much of our focus was on improving processes, with the enhancing the quality of fractal analysis.

References [1] Caserta, A., Consolini, G., & De Michelis, P. (2007). Statistical features of the seismic noise-field. Studia Geophysica et Geodaetica, 51(2), 255-266. [2] Matcharashvili, T., Chelidze, T., Javakhishvili, Z., Zhukova, N., Jorjiashvili, N., & Shengelia, I. (2013). Discrimination between stochastic dynamics patterns of ambient noises (Case study for Oni seismic station). Acta Geophysica, 61(6), 1659-1676. [3] Pilz, M., & Parolai, S. (2014). Statistical properties of the seismic noise field: influence of soil heterogeneities. Geophysical Journal International, 199(1), 430-440. [4] Wapenaar, K., Draganov, D., Snieder, R., Campman, X., & Verdel, A. (2010a). Tutorial on seismic interferometry: Part 1—Basic principles and applications. Geophysics, 75(5), 75A195-175A209. [5] Wapenaar, K., Slob, E., Snieder, R., & Curtis, A. (2010b). Tutorial on seismic interferometry: Part 2—Underlying theory and new advances. Geophysics, 75(5), 75A211-275A227.

===============================================================

Q (5) The title of subsection 3.1 seems not appropriate since the fractality of a signal can be detected or identified and not learned.

Answer: Thanks, we agree. This will be revised definitely.

===============================================================

Q (6) It is obscure at all, the summation the authors did in eq. 3; practically they sum over the three different time series, so instead to consider one set of fluctuation functions (depending on q) for each time series, they summed for each q the three fluctuation functions obtained for each time series. I suppose that each time series refers to one direction of the sensor by which the seismic noise was measured, one vertical and two horizontal. Actually, it would have been much more useful and informative of the underlying geophysical process to analyze each direction separately. Probably it would have been better to first calculate the total displacement combining the three-time series and then apply the MFDFA on such total displacement.

Answer: Actually, we followed a process similar to the one outlined in the "Casetra, et al., 2007" provided in Eq. (2). The similar approach has been taken by Pilz & Parolai (2014) in Eq. (2). The rationale behind this suggestion was given by Casetra, et al., (2007) as: "Moreover, we consider the 3D soil displacement instead of its three components because we are interested in studying the global soil motion under the effect of seismic noise; considering and comparing the motion in each component separately (H/V spectral ratio, etc.), could be done in a next paper (Casetra, et al., 2007, p. 259)". In any case, we feel that your comment is well-founded and that there needs to be the reflection on the matter in the advanced processing.

Q (7) Eq (6) is not correct because the zero-values are of the parabolic fitting function and not of Dq, calculated from the data.

Answer We exactly do the method introduced by Eq. (9) in Padhy (2016), or by Eq. (21) in Shimizu et al. (2002). Maybe we don't quite understand the question of the referee. We would appreciate if you could notify the problem of this method to us. Q (8) At page 7, the authors say that the "phase structure, which controls the non-stationarity. . ." this is not correct because the phase is responsible for the non-linearity of a time series. The stationarity/nonstationarity of a time series can be simply verified looking at the power spectrum and its power-law shape, which depends solely on the amplitude of the Fourier transform and not in the phase.

Answer I appreciate for pointing out the mistakes I have made. It will be revised.
=============================================================

Q (9) In response to questions and comments from referees concerning the TFSA, we try to restate the method in the clear way in the attached supplement.
=============================================================

Q (10) At page 9, N/6 samples correspond to 600 seconds and not 6 seconds. Answer Signals have been recorded at 50 sample per second so each one of these signals has N = 3600 × 50 = 180000 samples. By limiting the size of segments into the h ≤ N / 6 samples, that is, h ≤ 600 s. Sorry for this mistake in typing. The authors mention that they calculated the modified Fq to compensate the effect at small scales, but at the end, they showed in Fig. 3a the un-modified Fq. 3) Fig. 3a shows that the fluctuation functions by approximately two different regimes, at small scales until approximately and at long scales from 1 until the end of the investigated timescale range. The authors do not do any comment on such apparent double regimes and it seems that they have calculated the scope of the lines fittings each fluctuation function in the entire range of scales (actually, the authors have not clarified/specified if they used the entire scale range or part of it). If they used the entire range, the results

obtained about the multifractality are absolutely wrong. 4) It is lacking a convincing geophysical explanation of the link between the tectonics of the area and the found multifractal parameters.

Please also note the supplement to this comment:
https://www.nonlin-processes-geophys-discuss.net/npg-2017-16/npg-2017-16-AC2-supplement.pdf
* * *
[Figure]

**Fig. 1.** Fig. 1. Amplitude modulated (AM) at left and frequency modulated (FM) signal at right observed over different time intervals (shown at different rows) (Borgnat et al. 2010).

[Figure]

**Fig. 2.** Fig. (2)

---

## Author Comment (AC3) · 9 Jan 2018

The author would indeed prefer to answer these three questions together. Q (1) It is not clear (in the worst case, not correct) that the long-range correlation of a stationary time series (with $0 < h(2) < 1$) can be discerned from a non-stationary process. Stationarity and non-stationarity are different characteristics and such relationship that makes one be discerned from the other is obscure.

Q (2) Moreover, the authors have not clarified, or not mentioned at all, what type of

nonstationarities would affect their data, so that the application of the MFDFA directly would produce misleading results. If the nonstationarities of their data are among those types that MFDFA would be able to deal with, why the pre-processing is proposed in the next sections?

Q (3) It is not clear the concern of the authors in selecting the stationary intervals of signals before using the MFDFA, if the MFDFA is already capable to deal with nonstationarities. Moreover, the authors have not clarified, or not mentioned at all, what type of nonstationarities would affect their data, so that the application of the MFDFA directly would produce misleading results. If the nonstationarities of their data are among those types that MFDFA would be able to deal with, why the pre-processing is proposed in the next sections?

Answer:

In general, there is broad agreement on the appropriateness of MFDFA in studying multifractal scaling behavior of non-stationary time series, but we wish to draw your attention to the paragraph "III. ANALYSIS OF SUNSPOT TIME SERIES" of Movahed (et al., 2005) which explains the reasons why further attention must be taken in the analyzing the stationarity of signals in the pre-processing step, before feeding them into the cycle of Fractal analysis. In connection with this point, we MUST make it absolutely clear that inherent non-stationarity of signals should never be confused with the concept of non-stationarity made by external perturbations. Regularly, fractional Gaussian noises (fGn) are involved in the inherent stationarity process, in contrast, fractional Brownian motions (fBm) are linked to the inherently non-stationarity process. The stationarity of fGn signals can be characterized by two parameters, $\sigma^{\hat{}}2$, the variance, and H, the Hurst coefficient, while a fBm process has a time-dependent variance. Hence, on the basis of the class to which signals belong, different techniques may be required for processing. Failure to match signal class with the appropriate method of fractal analysis results in serious error in the estimating H. For instance Dispersional analysis (Disp) is recommended to the analysing the fractionality of fGn signals, while bridge detrended scaled windowed variance analysis (bdSWV) is suitable for fBm signals (Eke et al., 2000). These classes might not be a-priori known, so a preliminary interpretation in this regard may be available by fitting a straight line of slope $-\beta$ on a log-log plot of the periodogram. Based on this method, signals can be categorized according to the value of $\beta$, but Eke et al., (2000) placed emphasis on this point that this method is only applicable if $\beta$ falls into the category $-1<\beta<0.38$ (for an obvious stationary case) or if falls into the range of $1.4<\beta<3$ (for an obvious non-stationary case), but there is no certainty that this method fully comply with the complicated characteristics of signals in the range of $3.8<\beta<1.4$ where stationary and non-stationary mixed into each other. Based on this assumption, it is essential to provide another reliable framework for a regular monitoring of stationarity of signals. Signal summation conversion method (SSC) is advised to use as a discriminating method (Eke et al., 2000). An alternative approach for distinguishing fGn signals from fBm signal was proposed by Movahed et al., (2005) who experimentally attempted to examine the feature of Sunspot Time Series by analyzing the behavior of standard deviation of its time series as a function of timescale. There is a growing consensus amongst researchers (Zhong, et al., 2015; Wang et al., 2014) that Time-Frequency Surrogate Analysis (TFSA) proposed by Borgnat et al. (2010) can provide a complementary view for testing stationarity of seismic signals in an operational sense. This method is statistically characterized on the basis of a set of surrogates which all share the same average spectrum as the analyzed signal while being stationarized. With this short introduction, the answers to the above-mentioned question can be summarized briefly: We, therefore, agree with this statement in your question: Stationarity and non-stationarity are different characteristics, and this is exactly what we need. In other words, this difference enables us to discern a stationary fGn from non-stationary fBm. We want to underline the point that nonstationarities from our point of view are those corresponding to inherent non-stationarity which should be known at the pre-processing step before making our choice between fGn and fBm process. TFSA is a rigorous and sufficiently flexible method which not only provides an opportunity for assessing the inherent stationarity/non-stationarity of a signal but

also further it may bring information from their state of stationarity. Therefore, another strength of TFSA lies in its capability to constrain the length of stationary of the signal by quickly and reliably validating tasks. This would be of utmost importance if our data encompass the microseismic range of frequency (0.1-0.3 Hz) or if data were acquired from the surveys at the vicinity of dome-looking topography features seismic signals appear mostly in the quasi-stationary state. In those cases, reproducible seismic signals could fall into one of the following states: macroscale, mesoscale or microscale state (Fig 1), therefore, more accurate method is needed in order to properly assess the state of this quasi-stationarity. In this paper, we do make a point that the length of the signal directly impacts on the reliability of Long-range autocorrelations assessment. The importance of this factor was previously the subject of other investigations such as Delignieres et al., (2006) and Warlop et al., (2017), but this point has been hidden from view in analyzing the fractality of seismic signals. The importance of this issue is strikingly apparent for seismic signals since the stationarity of seismic records at various frequency ranges or temporal length are different. The experimental estimates obtained by Gorbatikov & Stepanova, (2008) shows that, at the microseismic range of frequency, signals are mostly quasi-stationarity, but this stationarity may not be preserved for very long periods of time. For instance, this interval might be lengthened to the several days or be shortened to the 1–1.5 h, while Wang et al., (2014) showed that for frequencies above than 1 Hz signals, the stationarity range of the signal is just in the range of several seconds. This is where we think we have to focus much of our attention. Obviously, by choosing an extreme short window length users might thereby misunderstand the true origin of non-stationarity, in such a way that lead to results that are non-informative, and potentially misleading. Accordingly, we will be able to adaptively adjust the length of processing incompatible with the stationarity length of the signal, if we take full benefit of the advantages of TFSA.

References

[1] Bashan, A., Bartsch, R., Kantelhardt, J. W., & Havlin, S. (2008). Comparison of

detrending methods for fluctuation analysis. Physica A: Statistical Mechanics and its Applications, 387(21), 5080-5090.

[2] Borgnat, P., Flandrin, P., Honeine, P., Richard, C., & Xiao, J. (2010). Testing stationarity with surrogates: A time-frequency approach. IEEE Transactions on Signal Processing, 58(7), 3459-3470. Eke, A., Herman, P., Bassingthwaighte, J., Raymond, G., Percival, D., Cannon, M., et al. (2000). Physiological time series: distinguishing fractal noises from motions. Pflügers Archiv, 439(4), 403-415.

[3] Movahed, M. S., Jafari, G., Ghasemi, F., Rahvar, S., & Tabar, M. R. R. (2006). Multifractal detrended fluctuation analysis of sunspot time series. Journal of Statistical Mechanics: Theory and Experiment, 2006(02), P02003.

[4] Wang, D., Li, Y., & Nie, P. (2014). A study on the Gaussianity and stationarity of the random noise in the seismic exploration. Journal of applied Geophysics, 109, 210-217.

[5] Warlop, T., Bollens, B., Detrembleur, C., Stoquart, G., Lejeune, T., & Crevecoeur, F. (2017). Impact of series length on statistical precision and sensitivity of autocorrelation assessment in human locomotion. Human Movement Science, 55, 31-42.

[6] Zhong, T., Li, Y., Wu, N., Nie, P., & Yang, B. (2015a). Statistical analysis of background noise in seismic prospecting. Geophysical Prospecting, 63(5), 1161-1174.

[7] Zhong, T., Li, Y., Wu, N., Nie, P., & Yang, B. (2015b). Statistical properties of the random noise in seismic data. Journal of applied Geophysics, 118, 84-91.

============================================================

Q (4) The authors state "Existence of a self-affine long-range persistence in the seismic noise time series evidence that the current state of system is not in the pure diffused regime and transition from coherent to incoherent motion is still on progress" however not in the Results section nor in the Discussion section it was ever strengthened such statement on the base of the obtained results, leaving it suspended and without a clear connection with all the performed analysis.

Answer:

Generally speaking, a local perturbation may continuously fluctuate system over time in a complex manner, such that consecutive cycles of a signal exhibits an interdependency spanning over long time intervals. Indeed, the existence of this type of long-range correlation (or self-affine long-range persistence) in the seismic noise wavefield is mostly associated with the existence of coherent signals (e.g. P, S, and Surface waves). Matcharashvili, et al. (2013) showed that the dynamical features of ambient noise undergo essential changes during preparation, and also after triggering the activity of strong local events. At distance far away from a seismic source, the generated waves bounce on several heterogeneities and gradually enter in the multiple scattering regimes (see Fig. (2)). Therefore, the diffused scattered coda waves overwhelm the direct wave, gradually. In this case, the strength of persistence of signal diminishes through time, due to the attenuation of the coherent wavefront. Therefore, the footprint of a perturbation should be tracked in the ambient noises and the existence of a self-affine long-range persistence in the seismic noise time series evidences that this system is still in the process of transition, that is, transition from a complete coherent state to a pure diffusive state. In this point of view, the scattering mean free time, $\tau$, indicates the characteristic time after which such transition is happing. The scattering mean free time is the key feature of a medium since it allows knowing in advance the degree of heterogeneity of that medium. A medium with low levels of heterogeneity is no longer considered as a candidate site for executing the advanced process such as seismic interferometry (see, for instance, see Wapenaar, 2012 a,b), since the existence of a diffusive wavefield is a key precondition for this analysis (Pilz & Parolai 2014). With this short introduction, the answers to the above-mentioned question can be summarized briefly: In short, this sentence "Existence of a self-affine long-range persistence in the seismic noise time series evidence that the current state of system is not in the pure diffused regime and transition from coherent to incoherent motion is still on progress" is evident from recent studies not directly the conclusion of our research but we MUST give the specific reference for this argue. Our paper is actually

the continuation of the effort made by Pilz & Parolai (2014), much of our focus was on improving processes, with the enhancing the quality of fractal analysis.

References [1] Caserta, A., Consolini, G., & De Michelis, P. (2007). Statistical features of the seismic noise-field. Studia Geophysica et Geodaetica, 51(2), 255-266.

[2] Matcharashvili, T., Chelidze, T., Javakhishvili, Z., Zhukova, N., Jorjiashvili, N., & Shengelia, I. (2013). Discrimination between stochastic dynamics patterns of ambient noises (Case study for Oni seismic station). Acta Geophysica, 61(6), 1659-1676.

[3] Pilz, M., & Parolai, S. (2014). Statistical properties of the seismic noise field: influence of soil heterogeneities. Geophysical Journal International, 199(1), 430-440.

[4] Wapenaar, K., Draganov, D., Snieder, R., Campman, X., & Verdel, A. (2010a). Tutorial on seismic interferometry: Part 1—Basic principles and applications. Geophysics, 75(5), 75A195-175A209.

[5] Wapenaar, K., Slob, E., Snieder, R., & Curtis, A. (2010b). Tutorial on seismic interferometry: Part 2—Underlying theory and new advances. Geophysics, 75(5), 75A211-275A227.

==========================================================

Q (5) The title of subsection 3.1 seems not appropriate since the fractality of a signal can be detected or identified and not learned.

Answer:

Thanks, we agree. This will be revised definitely.

==========================================================

Q (6) It is obscure at all, the summation the authors did in eq. 3; practically they sum over the three different time series, so instead to consider one set of fluctuation functions (depending on q) for each time series, they summed for each q the three fluctuation functions obtained for each time series. I suppose that each time series

refers to one direction of the sensor by which the seismic noise was measured, one vertical and two horizontal. Actually, it would have been much more useful and informative of the underlying geophysical process to analyze each direction separately. Probably it would have been better to first calculate the total displacement combining the three-time series and then apply the MFDFA on such total displacement.

Answer:

Actually, we followed a process similar to the one outlined in the "Casetra, et al., 2007" provided in Eq. (2). The similar approach has been taken by Pilz & Parolai (2014) in Eq. (2). The rationale behind this suggestion was given by Casetra, et al., (2007) as:

"Moreover, we consider the 3D soil displacement instead of its three components because we are interested in studying the global soil motion under the effect of seismic noise; considering and comparing the motion in each component separately (H/V spectral ratio, etc.), could be done in a next paper (Casetra, et al., 2007, p. 259)".

In any case, we feel that your comment is well-founded and that there needs to be the reflection on the matter in the advanced processing.

Q (7) Eq (6) is not correct because the zero-values are of the parabolic fitting function and not of Dq, calculated from the data.

Answer

We exactly do the method introduced by Eq. (9) in Padhy (2016), or by Eq. (21) in Shimizu et al. (2002). Maybe we don't quite understand the question of the referee. We would appreciate if you could notify the problem of this method to us.

Q (8) At page 7, the authors say that the "phase structure, which controls the non-stationarity. . ." this is not correct because the phase is responsible for the non-linearity of a time series. The stationarity/nonstationarity of a time series can be simply verified looking at the power spectrum and its power-law shape, which depends solely on the amplitude of the Fourier transform and not in the phase.

Answer

I appreciate for pointing out the mistakes I have made. It will be revised.
=================================================================

Q (9) In response to questions and comments from referees concerning the TFSA, we try to restate the method in the clear way in the attached supplement.
=================================================================

Q (10) At page 9, N/6 samples correspond to 600 seconds and not 6 seconds. Answer

Signals have been recorded at 50 sample per second so each one of these signals has N = 3600 × 50 = 180000 samples. By limiting the size of segments into the h ≤ N / 6 samples, that is, h ≤ 600 s. Sorry for this mistake in typing.

The authors mention that they calculated the modified Fq to compensate the effect at small scales, but at the end, they showed in Fig. 3a the un-modified Fq. 3) Fig. 3a shows that the fluctuation functions by approximately two different regimes, at small scales until approximately and at long scales from 1 until the end of the investigated timescale range. The authors do not do any comment on such apparent double regimes and it seems that they have calculated the scope of the lines fittings each fluctuation function in the entire range of scales (actually, the authors have not clarified/specified if they used the entire scale range or part of it). If they used the entire range, the results obtained about the multufractality are absolutely wrong. 4) It is lacking a convincing geophysical explanation of the link between the tectonics of the area and the found multifractal parameters.

Please also note the supplement to this comment:
https://www.nonlin-processes-geophys-discuss.net/npg-2017-16/npg-2017-16-AC3-supplement.pdf
2017-16, 2017.

**Supplement:**

A signal is stationary over a given observation scale if its spectrum undergoes no evolution in that scale. This assumption leads Bayram and Baraniuk, (2000) to use Multitaper Spectrograms (MS) for studying the time-dependent features of signals as

$$w_x(t,f) = \frac{1}{K}\sum_{k=0}^{K}\left|\int x(\tau)h_k(-t)e^{-j2\pi f\tau}d\tau\right|^2 \tag{1}$$

where $\{h_k(\tau - t), k = 1, \ldots, K\}$ stands for the first $K$ Hermite functions, which are used as the short-length windows. Bayram and Baraniuk (2000) used the Hermite functions $h_k^H(t)$ as the sliding windows since they give the best time-frequency localization and orthonormality in the time-frequency domain. Hermite functions can be obtained recursively, as follows

$$h_k^H(t) = \pi^{\frac{-1}{4}}\left(2^k k!\right)^{\frac{-1}{2}} e^{\frac{-t^2}{2}} H_k(t) \tag{2}$$

where $\{H_k(t), t \in N\}$ represents Hermite polynomials, defined by

$$H_k(t) = 2tH_{k-1}(t) - 2(k-2)H_{k-1}(t) \tag{3}$$

in which $H_0(t) = 1$ and $H_1(t) = 2t$. These family of windows are mutually orthonormal with elliptic symmetry and maximum concentration in the time-frequency domain. To define the global spectrum of signal, we should take the average of MS as (Xiao et al., 2007)

$$\langle w_x(t,f)\rangle_N = \frac{1}{N}\sum_{t=0}^{N} w_x(t,f) \tag{4}$$

For a stationary signal $w_x(t,f)/w_x^{av}(t,f)$ remains almost unchanged at the whole recording window, but in practice fluctuations in this ratio is inevitable. These fluctuations can be defined by a dissimilarity function as

$$c_t^x = D\big(w_x(t,f), w_x^{av}(t,f)\big), \; t = 0, \ldots, N \tag{5}$$

The significance of fluctuations can also be assessed by using surrogates (Borgnant et al., 2010). A surrogate is artificially produced in such a way that mimics statistical properties of real data. Isospectral surrogates have identical power spectra as the real signal but with randomized phases (Theiler et al., 1992). Once a collection of $J$ synthesized isospectral surrogates, $\{s_j(t), j = 1, \ldots, J\}$, are generated, the dissimilarity between local, $w_{s_j}(t,f)$, and global spectra, $w_{s_j}^{av}(t,f)$, for surrogates can be evaluated by (Borgnant et al., 2010)

$$\left\{c_t^{s_j} = D\left(w_{s_j}(t,f), w_{s_j}^{av}(t,f)\right), \; t = 0, \ldots, N, \; j = 1, \ldots, J\right\} \tag{6}$$

Borgnat et al., (2010) merged the Kullback-Leibler distance,

$$D_{KL}(A,B) = \int_\Omega \big(A(f) - B(f)\big)\log(A(f)/B(f))\,df \tag{7}$$

and log-spectral distance, $D_{KL}(A,B)$,

$$D_{LSD}(A,B) = \int_\Omega |\log(A(f)/B(f))|\,df \tag{8}$$

in the following combined form

$$D(A,B) = D_{KL}(A,B).\left(1 + D_{LSD}\left(\tilde{A}, \tilde{B}\right)\right) \tag{9}$$

In these equations $A$ and $B$ are two positive distributions and $\tilde{A}$ and $\tilde{B}$ indicate their normalized versions to the unity over the domain. The dissimilarity function $D(A,B)$ enables us to differentiate an amplitude-modulated or frequency-modulated non-stationary signal from a stationary one. Statistical variance $\Theta_1 = var(c_n^x)_{n=1,\ldots,N}$ gives the variance of $c_n^x$s. Similarly, for each one of $J$ synthesized surrogates we can define a separate variance as

$$\left\{\Theta_0(j) = var\left(c_n^{s_j}\right)_{n=1,\ldots,N}, \; j = 1, \ldots, J\right\} \tag{10}$$

These $\Theta_0$s can be assumed as a set of realizations of Gamma probability distribution with the following description

$$P(x; a, b) = \frac{1}{b^a \psi(a)} x^{a-1} \exp\left(-\frac{x}{b}\right) \tag{11}$$

As a null hypothesis original signals is supposed to be stationary but if it violates the predefined threshold γ, null hypothesis is rejected and non-stationarity is assumed, that is

$$\mathcal{J}(x) = \begin{cases} 1 \ if \ \Theta_1 > \gamma: \ non-stationarity \\ \ \ 0 \ if \ \Theta_1 < \gamma: \ stationarity \end{cases} \tag{12}$$

The threshold value for $\gamma$ is considered as a confidence level of 95% for probability distribution under the maximum likelihood sense. By comparing $\Theta_1$ and the estimates of $\Theta_0$, one can define the degree of stationarity. Quantitatively, these difference can be evaluated by index of non-stationarity (INS) (Xiao et al., 2007):

$$\text{INS} = \sqrt{\Theta_1 / \frac{1}{J} \sum_{n=1}^{J} \Theta_0(j)} \tag{13}$$

Further, note the result of stationarity test depends on the window length of spectrogram, $T_n$. This dependence can be analyzed by the scale of non-stationarity (SNS). It informs us that in which one/ones of considered values for $T_n$ the given threshold in Eq. (10) has been exceeded (Xiao et al., 2007):

$$\text{SNS} = \frac{1}{T} \arg \max_{T_n} \{\text{INS}(T_n)\} \tag{14}$$

---

## Author Comment (AC4) · 18 Jan 2018

*Comments from Referees,*

**The author would indeed prefer to answer these three questions together.**

**Q (1) It is not clear (in the worst case, not correct) that the long-range correlation of a stationary time series (with 0 < h(2) < 1) can be discerned from a non-stationary process. Stationarity and non-stationarity are different characteristics and such relationship that makes one to be discerned from the other is obscure.**

**Q (2) Moreover, the authors have not clarified, or not mentioned at all, what type of nonstationarities would affect their data, so that the application of the MFDFA directly would produce misleading results. If the nonstationarities of their data are among those types that MFDFA would be able to deal with, why the pre-processing is proposed in the next sections?**

**Q (3) It is not clear the concern of the authors in selecting the stationary intervals of signals before using the MFDFA, if the MFDFA is already capable to deal with nonstationarities. Moreover, the authors have not clarified, or not mentioned at all, what type of nonstationarities would affect their data, so that the application of the MFDFA directly would produce misleading results. If the nonstationarities of their data are among those types that MFDFA would be able to deal with, why the pre-processing is proposed in the next sections?**

*Answer:*

In general, there is broad agreement on the **appropriateness** of **MFDFA** in studying multifractal scaling behaviour of **non-stationary** time series, but we wish to draw your attention to the paragraph **"III. ANALYSIS OF SUNSPOT TIME SERIES"** of **Movahed (et al., 2005**) which explains the reasons why further attention must be taken in the analysing the stationarity of signals in the **pre-processing step**, before feeding them into the cycle of Fractal analysis. In connection with this point, we **MUST** make it absolutely clear that inherent non-stationarity of signals should **never** be confused with the concept of non-stationarity made by external perturbations. Regularly, **fractional Gaussian noises** (fGn) are involved in the **inherently** stationarity process, in contrast, **fractional Brownian motions** (fBm) are linked to the **inherently** non-stationarity process. The stationarity of fGn signals can be characterized by two parameters, $\sigma^2$, the variance, and H, the Hurst coefficient, while a fBm process has a time dependent variance. Hence, on the basis of the class to which signals belong, different techniques may be required for processing. Failure to match signal class with the appropriate method of fractal analysis results in **serious error** in the estimating H. For instance Dispersional analysis (Disp) is recommended to the analysing the fractionality of fGn signals, while bridge detrended scaled windowed variance analysis (bdSWV) is suitable for fBm signals (Eke et al., 2000). These classes might not be **a-priori known,** so a preliminary interpretation in this regard may be available by **fitting a straight line** of slope $-\beta$ on a log–log plot of the periodogram. Based on this method, signals can be categorized according to the **value** of $\beta$, but Eke et al., (2000) placed emphasis on this point that this method is only applicable if $\beta$ falls into the category $-1<\beta<0.38$ (for an obvious stationary case) or if falls into the range of $1.4 <\beta<3$ (for an obvious non-stationary case), but there is no certainty that this method fully comply with the complicated characteristics of signals in the range of $3.8<\beta<1.4$ where stationary and non-stationary **mixed into each other**. Based on this assumption, it is essential to provide another reliable framework for a

regular monitoring of stationarity of signals. Signal summation conversion method (SSC) is advised to use as a discriminating method (Eke et al., 2000). An alternative approach for distinguishing fGn signals from fBm signal was proposed by Movahed et al., (2005) who experimentally attempted to examine the feature of Sunspot Time Series by analysing the behaviour of standard deviation of its time series as a function of time scale. There is a growing consensus amongst researchers (Zhong, et al., 2015; Wang et al., 2014) that **Time-Frequency Surrogate Analysis (TFSA)** proposed by Borgnat et al. (2010) can provide a complementary view for testing stationarity of seismic signals in an operational sense. This method is statistically characterized on the basis of a set of **surrogates** which all share the same average spectrum as the analysed signal while being stationarized. With this short introduction, the answers to above-mentioned question can be summarized briefly: We therefore agree with this statement in your question: **Stationarity and non-stationarity are different characteristics**, and this is exactly what we need. In other words, this difference enables us to discern a stationary **fGn** from non-stationary **fBm.** We want to underline the point that nonstationarities from our point of view are those corresponding to inherent non-stationarity which should be known at the pre-processing step before making our choice between fGn and fBm process. TFSA is a rigorous and sufficiently flexible method which not only provides an opportunity for assessing the inherent stationarity/non-stationarity of a signal, but also further it may bring information from their **state** of stationarity. Therefore, another strength of TFSA lies in its capability to constrain the length of stationary of signal by quickly and reliably validating tasks. This would be of utmost importance if our data encompass the microseismic range of frequency (0.1-0.3 Hz) or if data were acquired from the surveys at the vicinity of dome-looking topography features seismic signals appear mostly in the **quasi-stationary** state. In those cases, reproducible seismic signals could fall into the one of the following states: **macroscale**, **mesoscale** or **microscale** state (Fig 1), therefore, more accurate method is needed in order to properly assess the state of this **quasi-stationarity.** In this paper, we do make a point that the **length** of signal directly impacts on the reliability of Long-range autocorrelations assessment. The importance of this factor was previously the subject of other investigations such as Delignieres et al., (2006) and Warlop et al., (2017), but this point has been hidden from view in analysing the fractality of seismic signals. The importance of this issue is strikingly apparent for seismic signals, since the **stationarity** of seismic records at various frequency ranges or temporal **length** are different. The experimental estimates obtained by Gorbatikov & Stepanova, (2008) shows that, at the microseismic range of frequency, signals are mostly **quasi-stationarity,** but this stationarity may not be preserved for very long periods of time. For instance, this interval might be lengthen to the several day or be shorten to **the 1–1.5 h** , while Wang et al., (2014) showed that for frequencies above than 1 Hz signals, the stationarity range of signal is just in the range of **several seconds**. This is where we think we have to focus much of our attention. Obviously, by choosing an extreme short window length users might thereby misunderstand the true origin of non-stationarity, in such a way that lead to results that are non-informative, and potentially misleading. Accordingly, we will be able to adaptively adjust the length of processing in compatible with the stationarity length of signal, if we take full benefit of the advantages of **TFSA.**

[Figure]

**Fig. 1.** Amplitude modulated (AM) at left and frequency modulated (FM) signal at right observed over different time intervals (shown at different rows) (Borgnat et al. 2010).

**References**

[1] Bashan, A., Bartsch, R., Kantelhardt, J. W., & Havlin, S. (2008). Comparison of detrending methods for fluctuation analysis. *Physica A: Statistical Mechanics and its Applications, 387*(21), 5080-5090.

[2] Borgnat, P., Flandrin, P., Honeine, P., Richard, C., & Xiao, J. (2010). Testing stationarity with surrogates: A time-frequency approach. *IEEE Transactions on Signal Processing, 58*(7), 3459-3470.

Eke, A., Herman, P., Bassingthwaighte, J., Raymond, G., Percival, D., Cannon, M., et al. (2000). Physiological time series: distinguishing fractal noises from motions. *Pflügers Archiv, 439*(4), 403-415.

[3] Movahed, M. S., Jafari, G., Ghasemi, F., Rahvar, S., & Tabar, M. R. R. (2006). Multifractal detrended fluctuation analysis of sunspot time series. *Journal of Statistical Mechanics: Theory and Experiment, 2006*(02), P02003.

[4] Wang, D., Li, Y., & Nie, P. (2014). A study on the Gaussianity and stationarity of the random noise in the seismic exploration. *Journal of applied Geophysics, 109*, 210-217.

[5] Warlop, T., Bollens, B., Detrembleur, C., Stoquart, G., Lejeune, T., & Crevecoeur, F. (2017). Impact of series length on statistical precision and sensitivity of autocorrelation assessment in human locomotion. *Human Movement Science, 55*, 31-42.

[6] Zhong, T., Li, Y., Wu, N., Nie, P., & Yang, B. (2015a). Statistical analysis of background noise in seismic prospecting. *Geophysical Prospecting, 63*(5), 1161-1174.

[7] Zhong, T., Li, Y., Wu, N., Nie, P., & Yang, B. (2015b). Statistical properties of the random noise in seismic data. *Journal of applied Geophysics, 118*, 84-91.

====================================================================

**Comments from Referees**

**Q (4) The authors state "Existence of a self-affine long range persistence in the seismic noise time series evidences that the current state of system is not in the pure diffused regime and transition from coherent to incoherent motion is still on progress" however not in the Results section nor in the Discussion section it was ever strengthened such**

**statement on the base of the obtained results, leaving it suspended and without a clear connection with all the performed analysis.**
*Answer:*

Generally speaking, a local perturbation may continuously fluctuate system over time in a complex manner, such that consecutive cycles of a signal exhibit an **interdependency** spanning over long time intervals. Indeed, existence of this type of **long range correlation** (or **self-affine long range persistence**) in the seismic noise wavefield is mostly associated with the existence of **coherent** signals (e.g. P, S, and Surface waves). Matcharashvili, et al. (2013) showed that the **dynamical** features of ambient noise undergo essential changes during preparation**,** and also after triggering the activity of strong local events. At distance far away from a seismic source, the generated waves bounce on several heterogeneities and gradually enter in the **multiple scattering regime** (see Fig. (2)). Therefore, the diffused scattered coda waves overwhelm the direct wave, gradually. In this case, the strength of persistence of signal diminishes through time, due to the attenuation of the coherent wave front. Therefore, the **footprint of a** perturbation should be tracked in **the ambient noises** and the **existence of a self-affine long range persistence** in the seismic noise time series evidences that this system is still in the process of transition, that is, transition from a complete coherent state to a pure diffusive state. In this point of view, the **scattering mean free time,** τ, indicates **the characteristic time** after which such transition is happing. The **scattering mean free time** is the key feature of a medium since it allows knowing in advance the **degree of heterogeneity** of that medium. A medium with low levels of heterogeneity is no longer considered as a **candidate site** for executing advanced process such as seismic interferometry (see for instance, see Wapenaar, 2012 a,b), since the existence of a diffusive wavefield is key precondition for this analysis (Pilz & Parolai 2014). With this short introduction, the answers to above-mentioned question can be summarized briefly:

In short, this sentence "Existence of a self-affine long range persistence in the seismic noise time series evidences that the current state of system is not in the pure diffused regime and transition from coherent to incoherent motion is still on progress" is **evident** from recent studies not directly the conclusion of our research but we **MUST** give the specific reference for this argue. Our paper is actually the continuation of the effort made by Pilz & Parolai (2014), much of our focus was on improving processes, with the enhancing the quality of fractal analysis.

[Figure]

Fig. (2)

*References*

 [1] Caserta, A., Consolini, G., & De Michelis, P. (2007). Statistical features of the seismic noise-field. *Studia Geophysica et Geodaetica, 51*(2), 255-266.

[2] Matcharashvili, T., Chelidze, T., Javakhishvili, Z., Zhukova, N., Jorjiashvili, N., & Shengelia, I. (2013). Discrimination between stochastic dynamics patterns of ambient noises (Case study for Oni seismic station). *Acta Geophysica, 61*(6), 1659-1676.

[3] Pilz, M., & Parolai, S. (2014). Statistical properties of the seismic noise field: influence of soil heterogeneities. *Geophysical Journal International, 199*(1), 430-440.

[4] Wapenaar, K., Draganov, D., Snieder, R., Campman, X., & Verdel, A. (2010a). Tutorial on seismic interferometry: Part 1—Basic principles and applications. *Geophysics, 75*(5), 75A195-175A209.

[5] Wapenaar, K., Slob, E., Snieder, R., & Curtis, A. (2010b). Tutorial on seismic interferometry: Part 2—Underlying theory and new advances. *Geophysics, 75*(5), 75A211-275A227.

=====================================================================

**Comments from Referees,**

**Q (5) The title of subsection 3.1 seems not appropriate, since the fractality of a signal can be detected or identified and not learned.**
*Answer:*

Thanks, we agree. This will be revised definitely.

**Comments from Referees**

**Q (6) It is obscure at all, the summation the authors did in eq. 3; practically they sum over the three different time series, so instead to consider one set of fluctuation functions (depending on q) for each time series, they summed for each q the three fluctuation functions obtained for each time series. I suppose that each time series refers to one direction of the sensor by which the seismic noise was measured, one vertical and two horizontal. Actually, it would has been much more useful and informative of the underlying geophysical process to analyse each direction separately. Probably it would has been better to first calculate the total displacement combining the three time series and then apply the MFDFA on such total displacement.**

*Answer:*

Actually, we followed a process similar to the one outlined in the "Casetra, et al., 2007" provided in Eq. (2). The similar approach has been taken by Pilz & Parolai (2014) in Eq. (2). The rationale behind this suggestion was given by Casetra, et al., (2007) as:

"*Moreover, we consider the 3D soil displacement instead of its three components because we are interested in studying the global soil motion under the effect of seismic noise; considering and comparing the motion in each component separately (H/V spectral ratio, etc.), could be done in a next paper* (Casetra, et al., 2007, p. 259)".

In any case, we feel that your comment is well-founded and that there needs to be reflection on the matter in the advanced processing.

**Q (7) Eq (6) is not correct, because the zero-values are of the parabolic fitting function and not of Dq, calculated from the data.**

*Answer*

We exactly do the method introduced by Eq. (9) in Padhy (2016), or by Eq. (21) in Shimizu et al. (2002). Maybe we don't quite understand the question of referee. We would appreciate if you could notify the problem of this method to us.

**Q (8) At page 7, the authors say that the "phase structure, which controls the non-stationarity. . ." this is not correct, because the phase are responsible of the non-linearity of a time series. The stationarity/nonstationarity of a time series can be simply verified looking at the power spectrum and its powerlaw shape, which depends solely on the amplitude of the Fourier transform and not on the phase.**

*Answer*

I appreciate for pointing out the mistakes I have made. It will be revised.

=================================================================

***Comments from Referees,***

**Q (9) In response to questions and comments from referees concerning the TFSA, we try to restate the method in the clear way as follows:**
 *Answer*

Maybe I was not quite clear enough in explaining the theoretical aspect of Testing Stationary of Signal, so was maybe not something one would want to do too
A signal is stationary over a given observation scale if its spectrum undergoes no evolution in that scale. This assumption leads Bayram and Baraniuk, (2000) to use Multitaper Spectrograms (MS) for studying the time-dependent features of signals as

$$w_x(t,f) = \frac{1}{K}\sum_{k=0}^{K}\left|\int x(\tau)h_k(-t)e^{-j2\pi f\tau}d\tau\right|^2 \tag{1}$$

where $\{h_k(\tau - t), k = 1, \dots, K\}$ stands for the first $K$ Hermite functions, which are used as the short-length windows. Bayram and Baraniuk (2000) used the Hermite functions $h_k^H(t)$ as the sliding windows since they give the best time-frequency localization and orthonormality in the time-frequency domain. Hermite functions can be obtained recursively, as follows

$$h_k^H(t) = \pi^{\frac{-1}{4}}\left(2^k k!\right)^{\frac{-1}{2}} e^{\frac{-t^2}{2}} H_k(t) \tag{2}$$

where $\{H_k(t), t \in N\}$ represents Hermite polynomials, defined by

$$H_k(t) = 2tH_{k-1}(t) - 2(k-2)H_{k-1}(t) \tag{3}$$

in which $H_0(t) = 1$ and $H_1(t) = 2t$. These family of windows are mutually orthonormal with elliptic symmetry and maximum concentration in the time-frequency domain. To define the global spectrum of signal, we should take the average of MS as (Xiao et al., 2007)

$$\langle w_x(t,f)\rangle_N = \frac{1}{N}\sum_{t=0}^{N} w_x(t,f) \tag{4}$$

For a stationary signal $w_x(t,f)/w_x^{av}(t,f)$ remains almost unchanged at the whole recording window, but in practice fluctuations in this ratio is inevitable. These fluctuations can be defined by a dissimilarity function as

$$c_t^x = D\left(w_x(t,f), w_x^{av}(t,f)\right), t = 0, \dots, N \tag{5}$$

The significance of fluctuations can also be assessed by using surrogates (Borgnant et al., 2010). A surrogate is artificially produced in such a way that mimics statistical properties of real data. Isospectral surrogates have identical power spectra as the real signal but with randomized phases (Theiler et al., 1992). Once a collection of $J$ synthesized isospectral surrogates, $\{s_j(t), j = 1, \ldots, J\}$, are generated, the dissimilarity between local, $w_{s_j}(t, f)$, and global spectra, $w_{s_j}^{av}(t, f)$, for surrogates can be evaluated by (Borgnant et al., 2010)

$$\left\{ c_t^{s_j} = D\left( w_{s_j}(t, f), w_{s_j}^{av}(t, f) \right), t = 0, \ldots, N, j = 1, \ldots, J \right\} \tag{6}$$

Borgnat et al., (2010) merged the Kullback-Leibler distance,

$$D_{KL}(A, B) = \int_{\Omega} \left( A(f) - B(f) \right) \log(A(f)/B(f)) \, df \tag{7}$$

and log-spectral distance, $D_{KL}(A, B)$,

$$D_{LSD}(A, B) = \int_{\Omega} |\log(A(f)/B(f))| \, df \tag{8}$$

in the following combined form

$$D(A, B) = D_{KL}(A, B) . \left( 1 + D_{LSD}(\tilde{A}, \tilde{B}) \right) \tag{9}$$

In these equations $A$ and $B$ are two positive distributions and $\tilde{A}$ and $\tilde{B}$ indicate their normalized versions to the unity over the domain. The dissimilarity function $D(A, B)$ enables us to differentiate an amplitude-modulated or frequency-modulated non-stationary signal from a stationary one. Statistical variance $\Theta_1 = var(c_n^x)_{n=1,\ldots,N}$ gives the variance of $c_n^x$s. Similarly, for each one of $J$ synthesized surrogates we can define a separate variance as

$$\left\{ \Theta_0(j) = var(c_n^{s_j})_{n=1,\ldots,N}, j = 1, \ldots, J \right\} \tag{10}$$

These $\Theta_0$s can be assumed as a set of realizations of Gamma probability distribution with the following description

$$P(x; a, b) = \frac{1}{b^a \psi(a)} x^{a-1} \exp\left( -x/b \right) \tag{11}$$

As a null hypothesis original signals is supposed to be stationary but if it violates the predefined threshold γ, null hypothesis is rejected and non-stationarity is assumed, that is

$$\mathcal{J}(x) = \begin{cases} 1 \text{ if } \Theta_1 > \gamma: \text{ non} - \text{stationarity} \\ 0 \text{ if } \Theta_1 < \gamma: \text{ stationarity} \end{cases}$$

(12)

The threshold value for $\gamma$ is considered as a confidence level of 95% for probability distribution under the maximum likelihood sense. By comparing $\Theta_1$ and the estimates of $\Theta_0$, one can define the degree of stationarity. Quantitatively, these difference can be evaluated by index of non-stationarity (INS) (Xiao et al., 2007):

$$\text{INS} = \sqrt{ \Theta_1 / \frac{1}{J} \sum_{n=1}^{J} \Theta_0(j) } \tag{13}$$

Further, note the result of stationarity test depends on the window length of spectrogram, $T_n$. This dependence can be analyzed by the scale of non-stationarity (SNS). It informs us that in which one/ones of considered values for $T_n$ the given threshold in Eq. (10) has been exceeded (Xiao et al., 2007):

$$\text{SNS} = \frac{1}{T} \arg \max_{T_n} \{ \text{INS}(T_n) \} \tag{14}$$

***References:***

Bayram, M. and Baraniuk, R.G., 2000. Multiple window time-varying spectrum estimation. *Nonlinear and Nonstationary signal processing*, pp.292-316.

Xiao, J., Borgnat, P., Flandrin, P. and Richard, C., 2007, August. Testing stationarity with surrogates- a one-class SVM approach. In *Statistical Signal Processing, 2007. SSP'07. IEEE/SP 14th Workshop on* (pp. 720-724). IEEE.

Borgnat, P., Flandrin, P., Honeine, P., Richard, C. and Xiao, J., 2010. Testing stationarity with surrogates: A time-frequency approach. *IEEE Transactions on Signal Processing*, *58*(7), pp.3459-3470.

=======================================================================

*Comments from Referees*

**Q (10) At page 9, N/6 samples correspond to 600 seconds and not 6 seconds.**
*Answer*
Signals have been recorded at 50 sample per second so each one of these signals has N = 3600 × 50 = 180000 samples. By limiting the size of segments into the h ≤ N ∕ 6 samples, that is, h ≤ 600 s. Sorry for this mistake in typing.

We also tested the process for different time length, different seasons, different weather conditions, and also night and day times. All of results will be added at the final paper. We confirmed the suitability of this method.

---

## Author Comment (AC6) · 22 Jan 2018

**EC1: The paper presents several flaws in the organization, in the methodological approach and in the presentation of the results. Unfortunately the English is not fluent and the clarity of the sentences and concepts is not always achieved. Even the abstract, which in principle has to outline clearly and synthetically the main findings of the study, seems quite obscure and does not convey clearly the information on what is the outcome of the presented research. For instance the starting paragraph of the Abstract "Existence of a self-affine long range persistence in the seismic noise time series evidences that the current state of system is not in the pure diffused regime and transition from coherent to incoherent motion is still on progress. Rate of this evolving transition can be indirectly linked to the degree of heterogeneity of medium" seems not well explained and it would be difficult for a reader to understand what exactly its meaning is. Please, be aware that at least the abstract should be developed in a manner that even a reader not strictly familiar with the topic of the paper can capture the general information. Unfortunately, the whole abstract fails in the characteristics of clarity, synthesis, clear explanation of the obtained results.**

**AC:** The extended abstract has been revised as follows. We hope it conveys clearly the main points of paper.

**Abstract.** The diffusivity of incoming seismic noise is certainly a critical precondition for executing seismic interferometry. But higher than the narrow ~ (0.05 -0.3) Hz microseismic bandwidth, this diffusivity stems mostly from the heterogeneity of local site characteristics, therefore the heterogeneity level of sites should be assessed beforehand in order to make an accurate assessment of a Green's response. As evidenced by recent studies (e.g. Padhy 2016), it has become evident that seismic signals show a self-affine long-range persistence in their coherent parts (e.g. P or S body waves) which is slowly disappeared with the emergence of the incoherent diffused incoming wavefield (i.e. Coda waves). Pilz & Parolai (2014) showed that the rate of this evolving transition is closely linked to the heterogeneity level of medium in such a way that for a strong heterogonous medium less time will be needed for falling signal into the diffuse state. Therefore, learning the fractality of a seismic noise will indirectly provide the basis for a decision on the potential place for executing seismic interferometry. But this conclusion rests on this pillar that input incoming noise wavefield is always stationary, but there is obviously a degree of ambiguity surrounding such assumption. There may be

circumstances under which signals include: Intrinsic Non-Stationary Direct Waves, Intrinsic Non-Stationary Scattered Waves and External Non-Stationary Signals. In executing the fractal analysis, it is essential that the method chosen be consistently reliable to ensure us that the correct Hurst coefficient is being used for the interpretation. There is broad agreement on the appropriateness of Multifractal Detrended Fluctuation Analysis (MF-DFA) in studying multifractal scaling behaviour of signals, corrupted by External Non-Stationary Signals, but it fails to comply with the intrinsic non-stationarity of signals. In this paper, we used the method introduced by Borgnat et al., (2010) to recognize the inherent characteristics of signal in the pre-processing step, before the feeding data into the cycle of Fractal analysis. Based on this revised method we try to define the degree of heterogeneity of different sites, locating at the North-Western of Iran.

References:

Borgnat, P., Flandrin, P., Honeine, P., Richard, C., and Xiao, J.: Testing stationarity with surrogates: A time-frequency approach, IEEE Transactions on Signal Processing, 58, 3459-3470, 2010.

Hillers, G., and Ben-Zion, Y.: Seasonal variations of observed noise amplitudes at 2–18 Hz in southern California, Geophysical Journal International, 184, 860-868, 2011.

Margerin, L., Planès, T., Mayor, J., and Calvet, M.: Sensitivity kernels for coda-wave interferometry and scattering tomography: theory and numerical evaluation in two-dimensional anisotropically scattering media, Geophysical Journal International, 204, 650-666, 2015.

Padhy, S.: The Multi-fractal Scaling Behavior of Seismograms Based on the Detrended Fluctuation Analysis, in: Fractal Solutions for Understanding Complex Systems in Earth Sciences, Springer, 99-115, 2016.

Pilz, M., and Parolai, S.: Statistical properties of the seismic noise field: influence of soil heterogeneities, Geophysical Journal International, 199, 430-440, 2014.

==============================================================================

**EC1: However, the authors say that after removing mean and trend (which trend? linear trend? a figure with the raw data would have been useful), they merged all the different length segments; but how such merging was performed? Then since the data present gaps "stemmed from the zeroed out spikes and overlaps" (what overlaps?), this gaps were filled with linear interpolation; but this interpolation is not clearly explained, and the number and the length of gaps is not specified: these details would be important to mention especially in a journal like NPG, where a relevant focus is given on the methodological aspect of presented study.**

**AC:** This question is of utmost importance and we will definitely add the detail of data processing. Our datasets include different-length segments (they are less than several seconds. As seen in the top panel of Figure 1, the first and end samples of each segments are accompanied by small-length "glitches". Further, some segments are affected by unusual trends which shows little consistence with regard to the exception of before and after it. Therefore, using the absolute-running-mean normalization can surgically remove narrow data glitches and unusual trends of a special segment (the below panel of Figure 1). The length of sliding window should be adaptively selected as small as possible to preserve the overall long-range mean and correlation of time series.

Missing data caused by removing these glitches is replaced by interpolated data as described at the Obspy Official website

https://docs.obspy.org/packages/autogen/obspy.core.trace.Trace.__add__.html#obspy.core.trace.Trace.__add__

(Part 4, from Handling gaps section named: "Traces with gaps and given fill_value='interpolate' ").

The length of interpolated sample is just one-sample so they potentially cannot effect on the long-range correlations. According to the Chen et al., (2002), shown at its Fig (2)-page 4, effects of the ''cutting'' procedure on the scaling behavior of correlated signals is not considerable with less than 10% of the points removed.

[Figure]

**Fig (1)** Above: An example for Merged 24-hour length signal which accompanied by small-length "glitches". Below: It depicts the signal shown in top panel after removing the small length glitches and replacing them with interpolated signals.

*References:*

Chen, Z., Ivanov, P.C., Hu, K. and Stanley, H.E., 2002. Effect of nonstationarities on detrended fluctuation analysis. Physical Review E, 65(4), p.041107.

===============================================================================

**EC1: An explanatory table indicating name, geographic coordinates, elevation of Stations should be attached. Such lack of clarity is also evidenced in the description of the dataset. It is not mentioned how many stations have been analysed, although one can guess them from Fig. 1; but probably an explanatory table indicating name, geographic coordinates, elevation, and maybe some simple statistical characteristics, would have been useful to add to make the text clearer.**

**AC:** We asserted below table as Table 1.

| Station | Latitude (N˚) | Longitude (E˚) | Altitude (m) |
|---------|---------------|----------------|--------------|
| AZR | 37.678 | 45.984 | 2273 |
| BST | 37.701 | 46.889 | 2112 |
| HRS | 38.318 | 47.042 | 2137 |
| MRD | 38.713 | 45.702 | 2142 |
| SHB | 38.283 | 45.619 | 2290 |
| SRB | 37.825 | 47.663 | 1958 |
| TBZ | 38.235 | 46.15 | 1550 |

===============================================================================

**EC1: Some flaws also exist in the methodology. For instance it would have been more correct to link the persistence/antipersistence of a signal to the succession of the increments rather than of the signal values.**

**AC:** Thank you for notification. We will correct that mistake, at the original paper.

===============================================================================

**EC1: It is correct the observation of the referee about what the authors did, ignoring the small part of the signal at its end that remains out during the calculation of the fluctuation function, since in**

**most of the studies such small part at the end of the signal has not ignored but included recalculating the fluctuation function starting from the end of the signal.**

**AC:** Actually, we followed a process similar to the one outlined in the "Casetra, et al., 2007, and Pilz & Parolai (2014) ". Your comment is greatly appreciated in this regard, but the similar approach has been taken, geared towards improving the efficiency of their introduced method. We also considered the 3D soil displacement instead of its three components given by Casetra, et al., (2007) as:

*"Moreover, we consider the 3D soil displacement instead of its three components because we are interested in studying the global soil motion under the effect of seismic noise; considering and comparing the motion in each component separately (H/V spectral ratio, etc.), could be done in a next paper* (Casetra, et al., 2007, p. 259)".

In any case, we feel that your comment is well-founded and that there needs to be reflection on the matter in the advanced processing.

***References:***

Caserta, A., Consolini, G. and De Michelis, P., 2007. Statistical features of the seismic noise-field. *Studia Geophysica et Geodaetica*, *51*(2), pp.255-266.

Pilz, M. and Parolai, S., 2014. Statistical properties of the seismic noise field: influence of soil heterogeneities. *Geophysical Journal International*, *199*(1), pp.430-440.

============================================================================

**Also the use of the multitaper spectrogram (Borgnat et al., 2010) seems not correctly performed or at least not clearly carried out, raising issues on the correctness of the obtained results.**

**AC:** Maybe I was not quite clear enough in explaining the theoretical aspect of Testing Stationary of Signal, so was maybe not something one would want to do too

A signal is stationary over a given observation scale if its spectrum undergoes no evolution in that scale. This assumption leads Bayram and Baraniuk, (2000) to use Multitaper Spectrograms (MS) for studying the time-dependent features of signals as

$$w_x(t,f) = \frac{1}{K}\sum_{k=0}^{K}\left|\int x(\tau)h_k(-t)e^{-j2\pi f\tau}d\tau\right|^2 \tag{1}$$

where $\{h_k(\tau - t), k = 1, \dots, K\}$ stands for the first $K$ Hermite functions, which are used as the short-length windows. Bayram and Baraniuk (2000) used the Hermite functions $h_k^H(t)$ as

the sliding windows since they give the best time-frequency localization and orthonormality in the time-frequency domain. Hermite functions can be obtained recursively, as follows

$$h_k^H(t) = \pi^{\frac{-1}{4}} \left(2^k k!\right)^{\frac{-1}{2}} e^{\frac{-t^2}{2}} H_k(t) \tag{2}$$

where $\{H_k(t), t \in N\}$ represents Hermite polynomials, defined by

$$H_k(t) = 2tH_{k-1}(t) - 2(k-2)H_{k-1}(t) \tag{3}$$

in which $H_0(t) = 1$ and $H_1(t) = 2t$. These family of windows are mutually orthonormal with elliptic symmetry and maximum concentration in the time-frequency domain. To define the global spectrum of signal, we should take the average of MS as (Xiao et al., 2007)

$$\langle w_x(t,f)\rangle_N = \frac{1}{N} \sum_{t=0}^{N} w_x(t,f) \tag{4}$$

For a stationary signal $w_x(t,f)/w_x^{av}(t,f)$ remains almost unchanged at the whole recording window, but in practice fluctuations in this ratio is inevitable. These fluctuations can be defined by a dissimilarity function as

$$c_t^x = D\big(w_x(t,f), w_x^{av}(t,f)\big), \ t = 0, \dots, N \tag{5}$$

The significance of fluctuations can also be assessed by using surrogates (Borgnant et al., 2010). A surrogate is artificially produced in such a way that mimics statistical properties of real data. Isospectral surrogates have identical power spectra as the real signal but with randomized phases (Theiler et al., 1992). Once a collection of $J$ synthesized isospectral surrogates, $\{s_j(t), j = 1, \dots, J\}$, are generated, the dissimilarity between local, $w_{s_j}(t,f)$, and global spectra, $w_{s_j}^{av}(t,f)$, for surrogates can be evaluated by (Borgnant et al., 2010)

$$\left\{c_t^{s_j} = D\left(w_{s_j}(t,f), w_{s_j}^{av}(t,f)\right), \ t = 0, \dots, N, \ j = 1, \dots, J\right\} \tag{6}$$

Borgnat et al., (2010) merged the Kullback-Leibler distance,

$$D_{KL}(A,B) = \int_{\Omega} \big(A(f) - B(f)\big) \log(A(f)/B(f)) \, df \tag{7}$$

and log-spectral distance, $D_{KL}(A,B)$,

$$D_{LSD}(A,B) = \int_{\Omega} |\log(A(f)/B(f))| \, df \tag{8}$$

in the following combined form

$$D(A,B) = D_{KL}(A,B).\left(1 + D_{LSD}(\tilde{A},\tilde{B})\right) \tag{9}$$

In these equations $A$ and $B$ are two positive distributions and $\tilde{A}$ and $\tilde{B}$ indicate their normalized versions to the unity over the domain. The dissimilarity function $D(A,B)$ enables

us to differentiate an amplitude-modulated or frequency-modulated non-stationary signal from a stationary one. Statistical variance $\Theta_1 = var(c_n^x)_{n=1,...,N}$ gives the variance of $c_n^x$s. Similarly, for each one of $J$ synthesized surrogates we can define a separate variance as

$$\left\{ \Theta_0(j) = var\left(c_n^{s_j}\right)_{n=1,...,N}, \; j = 1, ..., J \right\} \tag{10}$$

These $\Theta_0$s can be assumed as a set of realizations of Gamma probability distribution with the following description

$$P(x; a, b) = \frac{1}{b^a \psi(a)} x^{a-1} \exp\left(-\frac{x}{b}\right) \tag{11}$$

As a null hypothesis original signals is supposed to be stationary but if it violates the predefined threshold γ, null hypothesis is rejected and non-stationarity is assumed, that is

$$J(x) = \begin{cases} 1 \; if \; \Theta_1 > \gamma: non - stationarity \\ 0 \; if \; \Theta_1 < \gamma: stationarity \end{cases} \tag{12}$$

The threshold value for $\gamma$ is considered as a confidence level of 95% for probability distribution under the maximum likelihood sense. By comparing $\Theta_1$ and the estimates of $\Theta_0$, one can define the degree of stationarity. Quantitatively, these difference can be evaluated by index of non-stationarity (INS) (Xiao et al., 2007):

$$INS = \sqrt{\Theta_1 / \frac{1}{J} \sum_{n=1}^{J} \Theta_0(j)} \tag{13}$$

Further, note the result of stationarity test depends on the window length of spectrogram, $T_n$. This dependence can be analyzed by the scale of non-stationarity (SNS). It informs us that in which one/ones of considered values for $T_n$ the given threshold in Eq. (10) has been exceeded (Xiao et al., 2007):

$$SNS = \frac{1}{T} \arg \max_{T_n} \{INS(T_n)\} \tag{14}$$

**References:**

Bayram, M. and Baraniuk, R.G., 2000. Multiple window time-varying spectrum estimation. *Nonlinear and Nonstationary signal processing*, pp.292-316.

Xiao, J., Borgnat, P., Flandrin, P. and Richard, C., 2007, August. Testing stationarity with surrogates-a one-class SVM approach. In *Statistical Signal Processing, 2007. SSP'07. IEEE/SP 14th Workshop on* (pp. 720-724). IEEE.

Borgnat, P., Flandrin, P., Honeine, P., Richard, C. and Xiao, J., 2010. Testing stationarity with surrogates: A time-frequency approach. *IEEE Transactions on Signal Processing*, 58(7), pp.3459-3470.

===========================================================================

**EC1: The authors apply a complex signal pre-processing for searching the stationary windows to apply MFDFA. Besides the logical observation of the referee that the algorithm of the MFDFA is already developed in a way to remove the non-stationarities (thus making probably quite unuseful or unnecessary that pre-processing), it would have been, instead, much more useful, to apply the MFDFA directly to the signals (as obtained after the procedure described in section 2) and then to such stationary segments (and thus, after the pre-processing) to check if any difference would have been existed and to see if an improvement would have been obtained in the results, especially in relationship with the geophysical implications.**

**AC:** The background seismic noise, at all moments and over different realizations, is generally assumed to be a temporarily-stationary process with certain second-order statistical properties e.g., mean value, variance and autocorrelation functions. Also, other sources of non-stationarity (e.g. segments with different properties, random outliers or spikes with different amplitudes, etc.) are mostly viewed as superimposed external transient signals. In general, there is broad agreement on the appropriateness of MFDFA to get a better handle on the detrimental effects of simple types of superimposed intermittent non-stationarities, associating with exterior long period trends, e.g. polynomial, sinusoidal, and power-law functions, but In connection with this point, we must make it absolutely clear that inherent non-stationarity of signals should never be confused with the concept of non-stationarity made by external perturbations. We wish to draw your attention to the paragraph "III. ANALYSIS OF SUNSPOT TIME SERIES" of Movahed (et al., 2005) which explains the reasons why further attention must be taken in the analysing the inherent stationarity of signals in the pre-processing step, before feeding them into the cycle of Fractal analysis. The importance of this issue is strikingly apparent for seismic signals, since a seismic time series inherently may lose their stationarity within finite time intervals and theses intervals at various frequency ranges or temporal length are different. The experimental estimates obtained by Gorbatikov & Stepanova, (2008) shows that, at the microseismic range of frequency, signals are mostly quasi-stationarity, but this stationarity may not be preserved for very long periods of time. For instance, this interval might be lengthen to the several day or be shorten to the 1–1.5 h , while Wang et al., (2014) showed that for frequencies above than 1 Hz signals, the stationarity range of signal is just in the range of several seconds. Therefore, the choice for signal length in executing MFDFA may appear of utmost importance, in such a way that, choosing an extreme short window length might lead to non-informative,

and potentially misleading results. In certain circumstances, reproducible non-Stationary ballistic waves e.g. the waves induced by the interaction of wind and topography, or the waves generated by the near surface micro cracks may make matters worse by inducing large variability in the characteristics of seismic noise signals. Furthermore, Meng, et al., (2015) showed that the incoherent coda waves might be intrinsically non-Stationary, as well. This might be due to the occurrence of multiple scattering in an instable perturbed medium which Margerin et al., (2016) named it "the active scattering". When the time series is stationary its behavior can be considered as a fractal Gaussian noise (fGn), while for a no-stationary signal the concept of fractal Brownian motion (fBn) should be used instead (Qian, 2003; Ge and Leung, 2013). The stationarity of fGn signals can be characterized by two parameters, $\sigma^2$, the variance, and H, the Hurst coefficient, while a fBm process has a time dependent variance. Not surprisingly, on the basis of the class to which signals belong, different techniques may be required for processing. Seismic time series appear occasionally in the quasi-stationary state. In those cases, reproducible seismic signals could fall into the one of the following states: macroscale, mesoscale or microscale state (Borgnat et al. 2010), therefore, more accurate method is needed in order to properly assess the state of this quasi-stationarity. Failure to match signal class with the appropriate method of fractal analysis results in serious error in the estimating H and an incorrect interpretation of stationarity/non-stationarity of signal lead to misleading results (Chen et al., 2002; Eke, et al., 2002; Movahed et al., 2006). To this end, the Dispersional analysis (Disp) is recommended to use for alysing the fractionality of fGn signals, while bridge detrended scaled windowed variance analysis (bdSWV) is suitable for fBm signals (Eke et al., 2000). MFDFA can also be used sepately for both fGn and fBn (see, Delignieres et al., 2006). Based on this explanations, We want to underline that the degree of stationarity of signal should be known at the pre-processing step before making our choice between fGn and fBm process. However, these classes might not be a-priori known, so signal summation conversion method (SSC) is advised to use as a discriminating method (Eke et al., 2000). Based on this approach, a preliminary interpretation may be available by fitting a straight line of slope $-\beta$ on a log–log plot of the periodogram. Based on this method, signals can be categorized into the fGn or fBn, according to the value of β. Eke et al., (2000) placed emphasis on this point that tthe periodogram is only applicable for differentiating if β falls into the category $-1<\beta<0.38$ (for an obvious stationary case) or if falls into the range of $1.4 <\beta<3$ (for an obvious non-stationary case), but there is no certainty

that this method fully comply with the complicated characteristics of signals in the range of 3.8<β<1.4 where stationary and non-stationary mixed into each other. Therefore, it is essential to provide another reliable framework for a regular monitoring the stationarity of signals. In this paper, we do make a point that the length of signal directly impacts on the reliability of Long-range autocorrelations assessment. The importance of this factor was previously the subject of other investigations such as Delignieres et al., (2006) and Warlop et al., (2017).

**References**

[1] Bashan, A., Bartsch, R., Kantelhardt, J. W., & Havlin, S. (2008). Comparison of detrending methods for fluctuation analysis. *Physica A: Statistical Mechanics and its Applications, 387*(21), 5080-5090.
[2] Borgnat, P., Flandrin, P., Honeine, P., Richard, C., & Xiao, J. (2010). Testing stationarity with surrogates: A time-frequency approach. *IEEE Transactions on Signal Processing, 58*(7), 3459-3470.
[3] Eke, A., Herman, P., Bassingthwaighte, J., Raymond, G., Percival, D., Cannon, M., et al. (2000). Physiological time series: distinguishing fractal noises from motions. *Pflügers Archiv, 439*(4), 403-415.
[4] Delignieres, D., Ramdani, S., Lemoine, L., Torre, K., Fortes, M. and Ninot, G., 2006. Fractal analyses for 'short'time series: a re-assessment of classical methods. *Journal of Mathematical Psychology*, *50*(6), pp.525-544.
[5] Movahed, M. S., Jafari, G., Ghasemi, F., Rahvar, S., & Tabar, M. R. R. (2006). Multifractal detrended fluctuation analysis of sunspot time series. *Journal of Statistical Mechanics: Theory and Experiment, 2006*(02), P02003.
[6] Wang, D., Li, Y., & Nie, P. (2014). A study on the Gaussianity and stationarity of the random noise in the seismic exploration. *Journal of applied Geophysics, 109*, 210-217.
[7] Warlop, T., Bollens, B., Detrembleur, C., Stoquart, G., Lejeune, T., & Crevecoeur, F. (2017). Impact of series length on statistical precision and sensitivity of autocorrelation assessment in human locomotion. *Human Movement Science, 55*, 31-42.
[8] Zhong, T., Li, Y., Wu, N., Nie, P., & Yang, B. (2015a). Statistical analysis of background noise in seismic prospecting. *Geophysical Prospecting, 63*(5), 1161-1174.
[9] Zhong, T., Li, Y., Wu, N., Nie, P., & Yang, B. (2015b). Statistical properties of the random noise in seismic data. *Journal of applied Geophysics, 118*, 84-91.

**AC: I am also skeptical about the obtained results, because it seems that the calculation of the slopes of the fluctuation functions in Fig. 3 was performed considering all the available shown scales; if so, this is clearly wrong, because the fluctuation functions for any q are not linear in log-log scales. So, if the geophysical interpretation of the results are based on such wrong calculations of the slopes**

**of the fluctuation functions, also all the geophysical implications, rather poorly described by the way, would be not convincing.**

**EC:** We tested the process for different time length, different seasons, different weather conditions, and also night and day times. All of results will be added at the final paper. We confirmed the suitability of this method.